# Analysis of properties of the 19 February 2018 volcanic eruption of Mount Sinabung in S5P/TROPOMI and HIMAWARIHimawari-8 satellite data

A.T.J. de Laat[1], M. Vazquez-Navarro[2], N. Theys[3], and P. Stammes[1]

[1]KNMI, de Bilt, 3731 GK, the Netherlands
[2]DLR, Oberpfaffenhofen, D-82234 Wessling, Germany (currently at EUMETSAT)
[3]Royal Belgian Institute for Space Aeronomy (BIRA-IASB), Brussels, 1180, Belgium

*Correspondence to*: Jos de Laat (laatdej@knmi.nl)

**Abstract.** This study presents an analysis of TROPOMI cloud heights as a proxy for volcanic plume heights in the presence of absorbing aerosols and sulfur dioxide for the 19 February 2018 eruption plume of the Sinabung volcano on Sumatra, Indonesia.

Comparison with CALIPSO satellite data shows that all three TROPOMI cloud height data products based on oxygen absorption which are considered here (FRESCO, ROCINN, O22CLD) provide volcanic ash cloud heights comparable to heights measured by CALIPSO for optically thick volcanic ash clouds. FRESCO and ROCINN heights are very similar with only differences for FRESCO cloud top heights above 14 km altitude. O22CLD cloud top heights unsurprisingly fall below those of FRESCO and ROCINN, as the O22CLD retrieval is less sensitive to cloud top heights above 10 km altitude. For optically thin volcanic ash clouds, *i.e.* when Earth's surface or clouds at lower altitudes shine through the volcanic ash cloud, retrieved heights fall below the volcanic ash cloud heights derived from CALIPSO data.

Evaluation of corresponding Himawari-8 geostationary InfraRed (IR) brightness temperature differences (ΔBT) - a signature for detection of volcanic ash clouds in geostationary satellite data and widely used as input for quantitative volcanic ash cloud retrievals - reveals that for this particular eruption the ΔBT volcanic ash signature changes to a ΔBT ice crystal signature for the part of the ash plume reaching the upper troposphere beyond 10 km altitude several hours after the start of the eruption and which TROPOMI clearly characterizes as volcanic ($SO_2 > 1$ DU and AAI > 4 or more conservatively $SO_2 > 10$). The presence of ice in volcanic ash clouds is known to prevent the detection of volcanic ash clouds based on broadband geostationary satellite data. TROPOMI does not suffer from this effect, and can provide valuable and accurate information about volcanic ash clouds and ash top heights in cases where commonly used geostationary IR measurements of volcanic ash clouds fail.

## 1 Introduction

Monitoring airborne volcanic ash is of crucial importance for aviation planning, as volcanic ash is an environmental hazard that can cause damage to avionics systems, abrasion of exposed airframe parts, engine damage, and even engine failure [Prata and Rose, 2015]. From early 1980s onwards there have been several well-documented damaging encounters of (jet) aircraft with volcanic ash clouds. Since then, aviation authorities have set up working groups and task forces to develop guidelines,

procedures, and rules, on what to do in case of known or predicted volcanic ash [i.e. ICAO, 2012]. The advance of satellite remote sensing techniques in the early 2000s allowed for real-time global monitoring of volcanic eruptions and airborne volcanic ash and sulfur dioxide ($SO_2$), like the Support to Aviation Control Service - SACS [http://sacs.aeronomie.be; Brenot

et al., 2014] or the NOAA/CIMSS Volcanic Cloud Monitoring platform [https://volcano.ssec.wisc.edu/]. Nevertheless, in 2010, an eruption of the Icelandic volcano Eyjafjallajökull resulted in the closure of most of the European air space, stranding more than 8.5 million people and profoundly affecting commerce [Alexander, 2013]. The total economic damage was estimated at 2.2 billion $US [Oxford Economics, 2010]. In the aftermath of the 2010 eruption of Eyjafjallajökull, aviation authorities were quick to realize that aviation guidelines for volcanic ash avoidance were too strict. Since then, guidelines have

been updated [ICAO, 2012], allowing for more flexibility for aircraft to maneuver around volcanic ash clouds and giving airliners more responsibility. Furthermore, it was also recommended to further develop global real-time volcanic eruption and volcanic ash cloud monitoring services. Ongoing programs by ICAO and WMO continue to work on improving volcanic ash cloud satellite data products that can be used for real-time monitoring of volcanic eruptions and volcanic ash clouds, as well as for tactical and strategic flight planning [ICAO, 2012; WMO SCOPE, 2015, 2018].

However, despite the clear need for constant monitoring of volcanic eruptions and volcanic ash clouds, and despite the availability of a wide variety of satellite remote sensing data products to meet that particular need, a centralized facility to access and analyze all available remote sensing data on volcanic eruptions and volcanic ash clouds is still lacking. This strongly hampers integration of that information into aviation operations. As a consequence, volcanic eruptions continue to pose a larger than necessary risk for aviation.

In order to fill this information gap, the European Union funded the EUNADICS-AV project by the European Union's Horizon 2020 research program for "Societal challenges - smart, green and integrated transport". The main objective of EUNADICS-AV is "to close the significant gap in European-wide data and information availability during airborne hazards". Volcanic ash clouds are one of those airborne hazards. An important aspect of EUNADICS-AV is to verify how well various satellite instrument are capable of monitoring volcanic eruptions and volcanic ash clouds, and how to integrate various satellite data

products on board a variety of satellites. This requires integrated analyses of volcanic ash clouds with the current suite of satellites and remote sensing data.

For more than a decade, satellite instruments such as SCIAMACHY, OMI, GOME-2, OMPS, AIRS, and IASI, have been used to monitor volcanic eruptions in support of aviation. Measurements of $SO_2$ and the absorbing aerosol index (AAI) are currently provided in near-real-time (within 3 hours after the satellite spectral measurements) to the aviation community via the SACS

web-portal, which builds on the TEMIS project, which in 2003 provided the first web-based service that allowed to browse and download atmospheric satellite data products, also funded by ESA.

On 13 October 2017, ESA successfully launched the TROPOMI instrument as the single payload of ESA's S5P satellite [Veefkind et al., 2012]. TROPOMI is a grating spectrometer that measures Earth reflected radiances in the ultraviolet (UV), visible, near infrared (NIR), and shortwave infrared (SWIR) parts of the spectrum, building on the legacy provided by the

satellite instruments OMI and SCIAMACHY. Already a few weeks after launch, TROPOMI started to provide promising high

spatial resolution measurements (down to $3.5 \times 7$ km$^2$) of SO$_2$, the AAI, and cloud heights from various retrieval algorithms (FRESCO, O22CLD, ROCINN).

Compared to its predecessors, TROPOMI provides measurements with a better signal-to-noise ratio and much better spatial resolution (factor 10 or more, depending on the satellite that is compared with). This allows for a much better and more detailed characterization of volcanic ash and SO$_2$ plumes. Furthermore, due to a better spatial resolution and better instrumental signal-to-noise, TROPOMI is expected to provide improved height retrievals of volcanic ash clouds and volcanic SO$_2$, important parameter monitoring purposes [WMO SCOPE, 2015].

On 19 February 2018, 08:53 local time, the Indonesian volcano Mount Sinabung on Sumatra generated a dark gray plume with a high volume of ash that quickly rose to an estimated 15-17 km above sea level, according to the Darwin Volcanic Ash Advisory Center (VAAC). Ash plumes were identified in satellite images, recorded by webcams and smartphones, and widely shared on social media, also because of the time of the eruption (early morning) and the clear skies at that time. The event was possibly the largest since the beginning of the current episode of unrest at Sinabung, which started in September 2013 [https://volcano.si.edu/volcano.cfm?vn=261080; Eruptive History].

Mount Sinabung is located in Karo Regency, North Sumatra Province (03°10′ North, 98°23.5′ East, with a height of 2460 m a.s.l. [Hendrasto et al., 2012; Primulyana et al. 2017; Smithsonian Institute, Global Volcanism Program, 2019]. The stratovolcano had been dormant for more than 1200 years before it became active again in 2010, and especially since 2013 small eruptions have occurred regularly.

The 19 February 2018 Sinabung eruption provides one of the first possibilities to study the quality of TROPOMI data for volcanic cloud monitoring, also because there was a fortunate overpass of the CALIOP instrument on the CALIPSO satellite. CALIPSO was part of the A-train constellation, which consists of several Earth-observing satellites that closely follow one another, crossing the equator in an ascending (northbound) direction at about 13:30 local solar time, within seconds to minutes of each other along the same or a very similar orbital "track". The TROPOMI equator crossing time is comparable to those of satellites in the A-train constellation. Note that after an orbital maneuver in September 2018, CALIPSO is not part of the A-train constellation anymore.

In this paper, we evaluate satellite measurements of the 19 February 2018 Sinabung eruption with a particular focus on determining volcanic ash cloud heights combining TROPOMI AAI data with TROPOMI cloud height data. We also characterize the volcanic eruption plume in TROPOMI data, as well as compare TROPOMI data with geostationary Himawari-8 satellite IR data that are widely used for volcanic ash cloud detection. TROPOMI-based volcanic ash cloud heights are also compared with measurements from the CALIPSO satellite overpass.

## 2. Data

### 2.1 TROPOMI AAI

The AAI is a well-established data product that has been produced for several different satellite instruments spanning a period of more than 30 years. The AAI was first calculated as a correction for the presence of aerosols in column ozone measurements made by the TOMS instruments [Herman et al., 1997; Torres et al., 1998], because it was observed that ozone values were too high in typical regions of aerosol emission and transport. The AAI is based on spectral contrast in the ultraviolet spectral range for a given wavelength pair, where the difference between the observed reflectance and the modelled clear-sky reflectance results in a residual value. When this residual is positive it indicates the presence of UV-absorbing aerosols, like dust, smoke, or volcanic ash. Clouds yield near-zero residual values and negative residual values can be indicative of the presence of non-absorbing aerosols (e.g. sulphate), as shown by sensitivity studies of the AAI [e.g. de Graaf et al., 2005, Penning de Vries et al., 2009]. Unlike satellite-based aerosol optical thickness measurements, the AAI can also be calculated in the presence of clouds, so that daily global coverage is possible. This is ideal for tracking the evolution of episodic aerosol plumes from dust outbreaks, volcanic eruptions, and biomass burning. For this study, we use the TROPOMI AAI data for the wavelength pair 340-380 nm. For more details about the TROPOMI AAI retrieval algorithm, see Stein-Zweers [2016].

### 2.2 TROPOMI $SO_2$

Since the late 1970s, a large number of UV-visible satellite instruments have been used for monitoring anthropogenic and volcanic $SO_2$ emissions. In some cases, operational $SO_2$ retrieval streams have also been developed aiming to deliver $SO_2$ vertical column densities (VCD) in near real-time (NRT), i.e. typically with a delay of less than 3 hours.

The TROPOMI $SO_2$ retrieval algorithm is based on the DOAS technique [BIRA, 2016; Theys et al., 2017]. In brief, the log-ratio of the observed UV-visible spectrum, of radiation backscattered from the atmosphere, and an observed reference spectrum (solar or earthshine spectrum) is used to derive a slant column density (SCD), which represents the $SO_2$ concentration integrated along the mean light path through the atmosphere. This is done by fitting absorption cross-sections of $SO_2$ to the measured reflectance in a given spectral interval. In a second step, SCDs are corrected for possible biases. Finally, the SCDs are converted into vertical columns by means of air mass factors (AMF) obtained from radiative transfer calculations, accounting for the viewing geometry, clouds, surface properties, total ozone, and $SO_2$ vertical profile shapes. The TROPOMI $SO_2$ data product provides four different $SO_2$ VCDs for different $SO_2$ vertical profile shapes, since they are not known at the time of the measurement. For this paper, we use the standard $SO_2$ VCD data product.

### 2.3 TROPOMI cloud information

TROPOMI provides information about cloud properties by use of oxygen absorption in either the $O_2A$-band around 760 nm or the $O_2$-$O_2$ band around 477 nm [Veefkind et al., 2016]. In this study, we use the TROPOMI operational ROCINN cloud height [Loyola et al., 2018; Cloud as Reflecting Boundaries or CRB model] and FRESCO cloud height [Wang et al., 2008,

Wang et al., 2012], both based on the O2A-band, as well as off-line cloud height from the O22CLD algorithm based using the $O_2$-$O_2$ band [Veefkind et al., 2016]. Note that TROPOMI operational cloud fractions are derived from the OCRA algorithm [Loyola et al., 2018]. Both the FRESCO cloud height and the $O_2$-$O_2$ cloud height are based on a Lambertian cloud model. Therefore, the retrieved cloud height is the cloud mid-level rather than the cloud top [Wang et al., 2008, Sneep et al., 2012].

Note that because the current TROPOMI surface albedo databases – which rely on OMI data - are not fully representative for the TROPOMI spatial resolution and/or wavelengths, which results in inaccurate or unrealistic cloud retrievals which are flagged as missing data. It is expected that in the coming years a surface albedo database will be developed based on the TROPOMI measurements itself, which should solve these retrieval artefacts.

## 2.4 Himawari-8 AHI

The Advanced Himawari-8 Imager (AHI) is a geostationary satellite imager with 16 broad-band spectral channels from the visible to IR portion of the electromagnetic spectrum between 0.46 μm and 13.3 μm. The sub-satellite spatial resolution of AHI is 1 km for all-but-one VIS channels and 2 km for IR channels. The Himawari-8 AHI is a multipurpose imager that provides full-disk scans of Earth every 10 minutes from a geostationary orbit at 140.7°E. The imagery can be used for a variety of applications, including general environmental monitoring (e.g. cloud-tracked winds) and numerical weather prediction

[Bessho et al. 2016]. For the detection of volcanic ash clouds, results from an ad-hoc version of the VADUGS algorithm are used [Graf et al., 2015]. The VADUGS algorithm is a neural-network based on a large number of radiative transfer simulations of geostationary IR brightness temperatures, and retrieves the column mass loading ($kg/m^2$) and the top altitude of volcanic ash clouds. VADUGS was initially developed for SEVIRI/MSG, it has been adapted to Himawari-8 for the purpose of this paper. VADUGS uses the 10.8-12.0 μm channel $\Delta BT$ for geostationary IR volcanic ash clouds retrieval algorithms. The use

of this particular $\Delta BT$ is common practice [Prata, 1989], with negative $\Delta BT$ potentially indicating volcanic ash, and positive $\Delta BTs$ indicative of the presence of liquid water or ice content [Pavolonis et al., 2006].

## 2.5 CALIOP

The CALIOP lidar on board of the CALIPSO platform delivers global cloud and aerosol information. The vertical resolution of atmospheric profiles is high with 30-300m, but the horizontal sampling is poor, as the satellite is in a low-altitude earth orbit

with a 16- day repeated cycle and the horizontal resolution is only 330 m to 5 km [Winker et al., 2007, 2009]. In this study, we use 532 nm total attenuated backscatter (TAB) data from one CALIPSO orbit (data version 3.40) in a qualitative approach, *i.e.* detection of cloud and aerosol layers and their heights. The TAB signal strength is color coded such that blues correspond to molecular scattering and weak aerosol scattering, aerosols generally show up as yellow/red/orange. Stronger cloud signals are plotted in gray scales, while weaker cloud returns are similar in strength to strong aerosol returns and coded in yellows and

reds. The TAB in sensitive to atmospheric particles: both water and ice droplets as well as various types of aerosols.

## 3. Results

### 3.1 Brief description of the spatiotemporal evolution of the volcanic ash cloud

The analysis of Himawari-8 AHI IR brightness temperatures and IR-based volcanic ash cloud heights from CIMSS (Supplementary Information SI figure S1) shows that 19 February 2018 Sinabung eruption consisted of two distinct components. The initial eruption quickly reached the upper tropical troposphere (14-16 km altitude), after which the volcanic ash cloud was transported in a north/northwesterly direction. These heights are consistent with results from the recently introduced new TROPOMI $SO_2$ height data product [Hedelt et al., 2019]. Approximately two hours after the start of the eruption the satellite data shows lower-altitude volcanic ash cloud $\Delta BT$ signatures (up to 6-8 km altitude) emerging from under the high altitude volcanic ash cloud at both the northwest and southeast end of the high altitude volcanic ash cloud. As these lower altitude plumes also move more or less in opposite direction, they more likely reflect remnants of surface pyroclastic flows and/or the eruption column collapse that are also seen in the time-lapse webcam video footage on the internet (https://youtu.be/v45J5BO_ge0).

### 3.2 TROPOMI

Figure 1A shows the TROPOMI FRESCO cloud height and ROCINN cloud pressure, along with the TROPOMI AAI, and the AAI = 0 contour and the $SO_2$ = 10 Dobson Unit (DU) contour, with TROPOMI measurements within the figure area made at approximately 06:25 UTC, and 4.5 hours after the start of the eruption. By then, the volcanic plume has dispersed over an area with an approximate diameter of 200 km, while some parts of the volcanic ash cloud have sufficiently thinned so that cumulus clouds lower down in the atmosphere can be identified in VIIRS imagery (see SI figure S2; note that TROPOMI flies in a so-called loose formation with VIIRS, with a temporal separation between both of less than 5 minutes). The AAI and $SO_2$ contours agree well with the cloud structure associated with the volcanic plume , indicating there has not been a spatial separation between volcanic ash and $SO_2$, which is known to sometimes happen in volcanic eruptions [Cooke et al., 2014; Moxnes et al., 2014; Prata et al., 2017]. Guided by the AAI and $SO_2$ contour lines, the ash cloud can be identified in the FRESCO cloud height and ROCINN cloud pressure – in particular for cloud tops above 10 km – as well as in the FRESCO and O22CLD scene pressures (figure 1B), but not in the FRESCO cloud fraction (figure 1B), probably because of light absorption by ash. Comparing the cloud height with the VIIRS reflectances (SI figure S2), the volcanic plume altitudes occur where the ash cloud is sufficiently optically thick to not show the underlying surface and clouds.

All cloud height products show the same spatial structure with the highest clouds in the northern half of the ash plume. The FRESCO and ROCINN cloud heights both consistently indicate cloud heights of 10 km or higher, the O22CLD cloud heights also reach 10 km but for fewer pixels and in general FRESCO and ROCINN cloud heights are higher than the O22CLD cloud heights (figure 1B). The O22CLD data product is based on absorption of the $O_2$-$O_2$ complex, and is less sensitive to high altitude clouds as concentrations of the $O_2$-$O_2$ complex decrease strongly above approximately 10 km altitude [Acarreta et al., 2004]. The O22CLD algorithm is therefore computationally limited to a maximum cloud top pressures of 150 hPa (~13) km.

FRESCO and ROCINN are based on absorption of $O_2$, whose concentrations decrease much slower above 10 km altitude. The FRESCO and ROCINN cloud heights can therefore be used up to approximately 17 km altitude (~100 hPa) [Wang et al.,

2012]. The lower cloud height of O22CLD vs FRESCO/ROCINN is thus most likely due to the lower sensitivity of O22CLD for high clouds. Differences between FRESCO and ROCINN for the volcanic plume appear less striking, most notably the lack of saturated pixels in ROCINN (greys in FRESCO), possible due to the neural network filling in the gaps with nearby cloud information or interpolating between cloud pixels. However, it appears that FRESCO cloud heights are higher for the northern half of the ash plume. FRESCO cloud heights exceed 12.5 km, which is approximately 200 hPa, ROCINN cloud

pressure does not appear to exceed 200 hPa.

### 3.3 CALIOP

Although the 19 February 2018 Sinabung eruption was small in spatial extent and rather short-lived, by mere accident there was a perfect overpass with the CALIOP instrument in the A-train constellation (see Figure 1). The CALIOP track goes straight through the core of the volcanic ash cloud and across the north-south gradient in cloud tops.

Figure 2 shows the CALIOP backscatter signal at 532 nm overlaid with the TROPOMI FRESCO cloud heights, which are color coded according to the corresponding AAI values. The CALIOP overpass time of this area is between 07:09:56 and 07:11:26 UTC, the TROPOMI overpass time is between 06:24:23 and 06:26:00 UTC, a time difference of approximately 45 minutes. The CALIOP data clearly shows a cloud/ash layer around 15 km altitude, but also two cloud/ash structures extending from the ground up to approximately 10 km altitude, with an increase in height going from south to north. There is also a layer

detected in CALIPSO at 18 km around 3°N, which likely is also volcanic as the Himawari-8 BT does not provide any indication of other high clouds while there are negative ΔBTs near the CALIPSO track at 3°N, indicative of the presence of volcanic ash. There is a good agreement between the location of enhanced TROPOMI AAI values, FRESCO cloud height, and the altitude of high backscatter signal in the CALIOP data. The maximum cloud height in FRESCO agrees with the maximum backscatter height in CALIOP between 4° and 5° latitude. Between 3° and 4° latitude, the agreement is poor as the FRESCO cloud height

fall right in between the CALIOP backscatter data between 13-18 km altitude and those close to the surface. The CALIOP data also suggests that backscatter signals between 3° and 4° latitude are weaker than between 4° and 5° latitude, which might indicate less dense ash or clouds. For a semi-transparent cloud/ash plume it could be expected that FRESCO cloud heights are lower than the actual height of the cloud/ash plume due the presence of bright clouds nearer to the surface. Note that CALIOP's own feature mask does not identify hardly any of these backscatter signals as aerosol (for CALIOP v4.10 an occasional cloud

pixel is flagged as aerosol, see Hedelt et al., [2019]): the high-altitude structures are flagged as regular clouds, and the below-cloud structure as "totally attenuated", even though clearly the attenuation is not complete. The lack of aerosol masking in the feature mask most likely is related to liquid water or ice contaminating the volcanic ash [Hedelt et al., 2019].

Figure 3 shows the corresponding cloud heights from the O22CLD and ROCINN algorithms. The ROCINN cloud height is very similar to the FRESCO cloud height ($R^2 = 0.98$ for FRESCO cloud heights between 0.5 and 14 km regardless of

corresponding AAI value). The only difference occurs for FRESCO cloud heights > 14 km where the ROCINN cloud height

appears to be nearly constant around 12 km or 200 hPa. For the O22CLD data the maximum heights are on average lower than the FRESCO/ROCINN cloud heights. The lower cloud height of the O22CLD product is likely related to the reduced sensitivity of O22CLD for clouds above approximately 10 km altitude. Nevertheless, all products clearly indicate volcanic cloud heights of 10 km and higher, with the largest heights between 4° and 5° latitude, consistent with the CALIOP observation that backscatter signals between 3° and 4° latitude are weaker than between 4° and 5° latitude.

Although the CALIOP overpass is perfect in space, the time difference between TROPOMI and CALIOP of approximately 45 minutes is not insignificant. It is therefore unlikely that TROPOMI and CALIOP ash layers and structures exactly match. The flow direction of the volcanic ash cloud was northwards, which means that CALIOP should also be displaced north compared to TROPOMI. A rough estimate of northward cloud motion based on the geostationary satellite data indicates that the displacement may be approximately 0.5°/hour, which makes it not unreasonable to assume that some of the discrepancies between TROPOMI and CALIOP could also be related to the differences in observation time. Furthermore, volcanic eruption plumes have their own dynamics, with for example pyroclastic flows near the surface which appear to travel partly in the opposite direction of the background flow. The eruption dynamics may thus have additional effects on the ash plume displacement, for which time series of the complete 3-dimensional view of the eruption plume would be preferred. The current available satellite data only provide a 2-dimensional view of the eruption plume from above (geostationary, Polar orbiting), with information about changes over time in case of the geostationary satellites and with some but limited information about cloud and aerosol height. CALIOP measurements only provide one 2-dimensional cross-section through the eruption plume, without any information about changes over time.

### 3.4 Himawari-8

The temporal evolution of the ash plume was further investigated using Himawari-8 geostationary IR observations. Figure 4 shows the Himawari-8 10.8-12.0 μm channel (ΔBT) as observed between 02:30 UTC and 07:30 UTC in hourly intervals, including the TROPOMI SO$_2$/AAI contours shown in Figure 1.

During the first few hours (02:30-03:30), the ash plume is clearly visible both in the ΔBTs (reddish colors) and cloud heights (whites). At 03:30 UTC, two distinct clouds have emerged with fairly negative ΔBTs: one associated also with a high cloud height (white cloud colors), and another one further south with much lower cloud heights, likely low-altitude outflow or pyroclastic flows (blue cloud colors). From 04:30 UTC onwards, a third region becomes visible with high cloud heights and large positive ΔBTs (purple), indicative of high ice clouds, which continues to grow and expand northward.

Figure 5 shows a comparison of TROPOMI AAI and SO$_2$ data with regridded Himawari-8 ΔBTs (panel A). When focusing on AAI and SO$_2$ values, it appears that larger ΔBT values occur for smaller AAI values (< 2) and SO$_2$ columns (< 10 DU). The largest positive ΔBT are associated with optically thicker/less transparent water and ice clouds (see also VIIRS imagery in the SI and comparison of TROPOMI with CALIPSO). The lack of larger AAI and SO$_2$ values for larger positive ΔBT values therefore may reflect some kind of shielding of the volcanic ash and SO$_2$ by the iced upper levels of the volcanic ash cloud.

SO$_2$ may have been converted into sulphate as the SO$_2$ depletion rate (e-folding time), which, although uncertain, has been estimated to be as small as 5-30 minutes [Oppenheimer et al., 1998; McGonigle et al., 2004], scavenged by ice [Rose et al., 2000], or via ice nucleation of volcanic ash particles [Durant et al., 2008]. For negative $\Delta$BTs – indicative of volcanic ash clouds – we also find little evidence of a distinctive relation between either the AAI and SO$_2$ with $\Delta$BTs. This may similarly reflect a shielding effect, as the largest aerosol concentrations are not associated with the largest possible $\Delta$BTs [*e.g.* Prata and Prata, 2012; Pavolonis et al., 2016].

The emergence of an IR ice/water cloud signature within the volcanic ash cloud is consistent with analysis of available video footage and pictures on social media that show signs of condensation within the ash clouds soon after the start of the eruption. This is indicative of a moist troposphere in this area, which is further supported by the widespread development of (late) afternoon thunderstorms on 19 February throughout Sumatra. The eruption thus caused an increase in high altitude water vapor, either by moisture contained in the eruption itself or by the rapid vertical motions within the eruption column. The results presented here support the notion that the IR volcanic ash cloud $\Delta$BT signature disappears when condensed water vapor or ice forms in a volcanic ash cloud, which are known to significantly hamper IR volcanic ash cloud retrievals [Francis et al., 2012; Pavolonis et al., 2015a, 2015b; Zhu et al., 2017].

## 4. Discussion and conclusions

Analysis of measurements from the polar orbiting TROPOMI satellite - with unprecedented spatial resolution and accuracy – of the volcanic eruption of Mount Sinabung on Sumatra on 19 February 2018, has revealed that the combination of TROPOMI AAI and TROPOMI SO$_2$ allows for accurate identification of the volcanic ash cloud  location. In addition, under the condition that the ash plume is sufficiently thick so that clouds and the Earth surface below the ash cloud are not visible, TROPOMI cloud heights also provide accurate information about the volcanic ash cloud heights. The TROPOMI FRESCO and ROCINN cloud heights agree with CALIOP cloud top measurements for optically thick volcanic ash clouds. However, there is a difference between FRESCO and ROCINN for very high FRESCO heights (> 12.5 km or approximately 200 hPa). This might indicate that the ROCINN neural network may not be that sufficiently trained on clouds beyond 12 km or 200 hPa. In passing we note that the unprecedented spatial resolution of TROPOMI allows for detection of much smaller eruptions than is currently possible with polar orbiting satellite instruments like OMPS, GOME-2, and OMI. Also note that it could be argued that it would be better to use the TROPOMI SO$_2$ 15 km data product, as 15 km is more consistent with the volcanic plume height. However, this 15 km data product assumes a "nice and tidy" SO$_2$ plume without any contamination, let alone the complexity of a fresh, optically very thick eruption plume and the presence of condensed water, in combination with indications of a shielding effect. Furthermore, the main focus of this paper is ash heights rather than SO$_2$, which is mostly used as a proxy for a volcanic plume, although investigating the accuracy and precision of satellite SO$_2$ VCD observations in fresh volcanic plumes would be valuable, in particular with soon to be launched geostationary hyperspectral satellites.

Comparison with CALIOP aerosol and cloud heights provides clear indications that ash height estimates using cloud heights and AAI values from UV/VIS satellites like TROPOMI may underestimate actual ash heights in case of semi-transparent volcanic ash clouds, especially in the presence of high concentrations of water vapour and for very high altitude volcanic ash clouds. For  volcanic ash clouds optically thin enough for light to pass through the TROPOMI cloud heights are a weighted mean of the ash height and heights of other clouds or the surface, and are therefore less useful for volcanic ash cloud height monitoring purposes. Some discrepancies between TROPOMI and CALIPSO may be related due to misalignment in observation times of both satellite instruments (~ 45 minutes). In addition, indications were found of shielding of volcanic ash by this ice/water near top of the volcanic ash cloud.

There are also clear indications in the geostationary IR data of the formation of water/ice near the top of the volcanic ash cloud. The analysis of geostationary satellite data for this particular case revealed that under conditions of volcanic ash mixed with ice of condensed water, the geostationary IR volcanic ash cloud $\Delta$BT signature is lost and geostationary volcanic ash cloud retrievals cannot identify crucial parts of the ash plume. It is worth mentioning that the temporal resolution inherent to the geostationary orbit allows the observation of the onset and evolution of the plume, even in adverse conditions for IR volcanic ash cloud retrieval algorithm.

Polar orbiting satellites like TROPOMI thus may be better able to detect volcanic ash when condensed ice/water is present in volcanic plumes, in particular when synergistically combining different satellite data products like the AAI and $SO_2$. Furthermore, for the present case study, large negative $\Delta$BTs appear not to be a good indicator of large AAI values (or large $SO_2$ columns). This is not surprising as highly negative $\Delta$BTs do not necessarily indicate large ash optical depth values [*e.g.* Prata and Prata, 2012; Pavolonis et al., 2016]. Our results therefore highlight that there is added value in combining IR $\Delta$BT with UV/VIS AAI and $SO_2$. Satellite measurements like those from TROPOMI measurements thus can add significant value to geostationary IR volcanic ash cloud retrievals. Furthermore, in case of sufficiently dense ash , the cloud height data products provide accurate volcanic ash cloud heights, an important piece of information for aviation. For semi-transparent volcanic ash clouds, where the cloud top height retrievals become sensitive to other reflective surfaces below the transparent volcanic ash clouds, detection of accurate volcanic ash cloud heights is limited.

Hence, for AAI values larger than 4, TROPOMI cloud heights can be used for determining aerosol heights, and in case also $SO_2$ is detected such measurements should be interpreted as also containing volcanic ash (column values > 1 DU [Theys et al., 2017]). For more conservative estimates $SO_2$ column values > 10 could be considered. This AAI threshold value of 4 may be conservative but ensures that the aerosol layer very likely is opaque, as generally the associated aerosol optical depth will be (very) large [de Graaf et al., 2005]. For the combination of UV/VIS cloud heights, AAI and $SO_2$ could also be used for other UV/VIS satellites like GOME-2, OMPS, and OMI. These results highlight the importance of the integrated use of multiple (satellite) data sources for the detection and characterization of volcanic ash clouds, in particular for aviation purposes. This has been recognized by the European Union and is being further developed within the H2020 project EUNADICS-AV (http://www.eunadics.eu).

## Acknowledgements

The authors thank the two anonymous referees for their thoughtful and valuable comments. This paper is supported by the European Union's Horizon 2020 research and innovation programme under grant agreement No 723986, project EUNADICS-AV (European Natural Airborne Disaster Information and Coordination System for AViation).

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

**Glossary**

|        |               |                                                                                      |
|--------|---------------|--------------------------------------------------------------------------------------|
|        | AAI           | - Absorbing Aerosol Index                                                            |
| 435    | AIRS          | - Atmospheric InfraRed Sounder                                                        |
|        | AMF           | - Air Mass Factor                                                                     |
|        | AHI           | - Advanced Himawari Imager                                                            |
|        | BIRA          | - Belgian Institute for Space Aeronomy                                                |
|        | ΔBT           | - Brightness Temperature Difference                                                   |
| 440    | CALIOP        | - Cloud-Aerosol Lidar with Orthogonal Polarization                                   |
|        | CALIPSO       | - Cloud-Aerosol Lidar and Infrared Pathfinder Satellite Observations                 |
|        | CIMSS         | - Cooperative Institute for Meteorological Satellite Studies                          |
|        | DOAS          | - Differential Optical Absorption Spectroscopy                                        |
|        | DU            | - Dobson Unit                                                                         |
| 445    | ESA           | - European Space Agency                                                               |
|        | EUNADICS-AV   | - European Natural Airborne Disaster Information and Coordination System for AViation |
|        | FRESCO        | - Fast Retrieval Scheme for Clouds from the Oxygen A-band                             |
|        | GOME-2        | - Global Ozone Monitoring Experiment 2                                                |
|        | ICAO          | - International Civil Aviation Organization                                           |
| 450    | IASI          | - Infrared atmospheric sounding interferometer                                        |
|        | IR            | - InfraRed                                                                            |
|        | NOAA          | - National Oceanic and Atmospheric Administration                                     |
|        | NRT           | - Near Real Time                                                                      |
|        | OCRA          | - Optical Cloud Recognition Algorithm                                                 |
| 455    | OMI           | - Ozone Monitoring Instrument                                                         |
|        | OMPS          | - Ozone Mapping Profiler Suite                                                        |
|        | O22CLD        | - $O_2$-$O_2$ cloud                                                                   |
|        | ROCINN        | - Retrieval Of Cloud Information using Neural Networks                                |
|        | SACS          | - Support for Aviation Control Service                                                |
| 460    | SCD           | - Slant Column Density                                                                |
|        | SCIAMACHY     | - SCanning Imaging Absorption SpectroMeter for Atmospheric ChartographY              |
|        | SCOPE         | - Sustained, Coordinated Processing of Environmental satellite data for nowcasting   |
|        | SI            | - Supplementary Information                                                           |
|        | $SO_2$        | - Sulfur dioxide                                                                      |
| 465    | SUOMI-NPP     | - Suomi National Polar-orbiting Partnership                                           |
|        | S5P           | - Sentinel-5 Precursor                                                                |

| | | |
|---|---|---|
| | TAB | - Total Attenuated Backscatter |
| | TEMIS | - Tropospheric Emission Monitoring Internet Service |
| | TOMS | - Total Ozone Mapping Spectrometer |
| 470 | TROPOMI | - TROPOspheric Monitoring Instrument |
| | UTC | - Universal Time Coordinate |
| | UV | - UltraViolet |
| | VAAC | - Volcanic Ash Advisory Center |
| | VADUGS | - Volcanic Ash Detection Using Geostationary Satellites. |
| 475 | VCD | - Vertical Column Density |
| | VIS | - Visible |
| | VIIRS | - Visible Infrared Imaging Radiometer Suite |
| | WMO | - World Meteorological Organization |

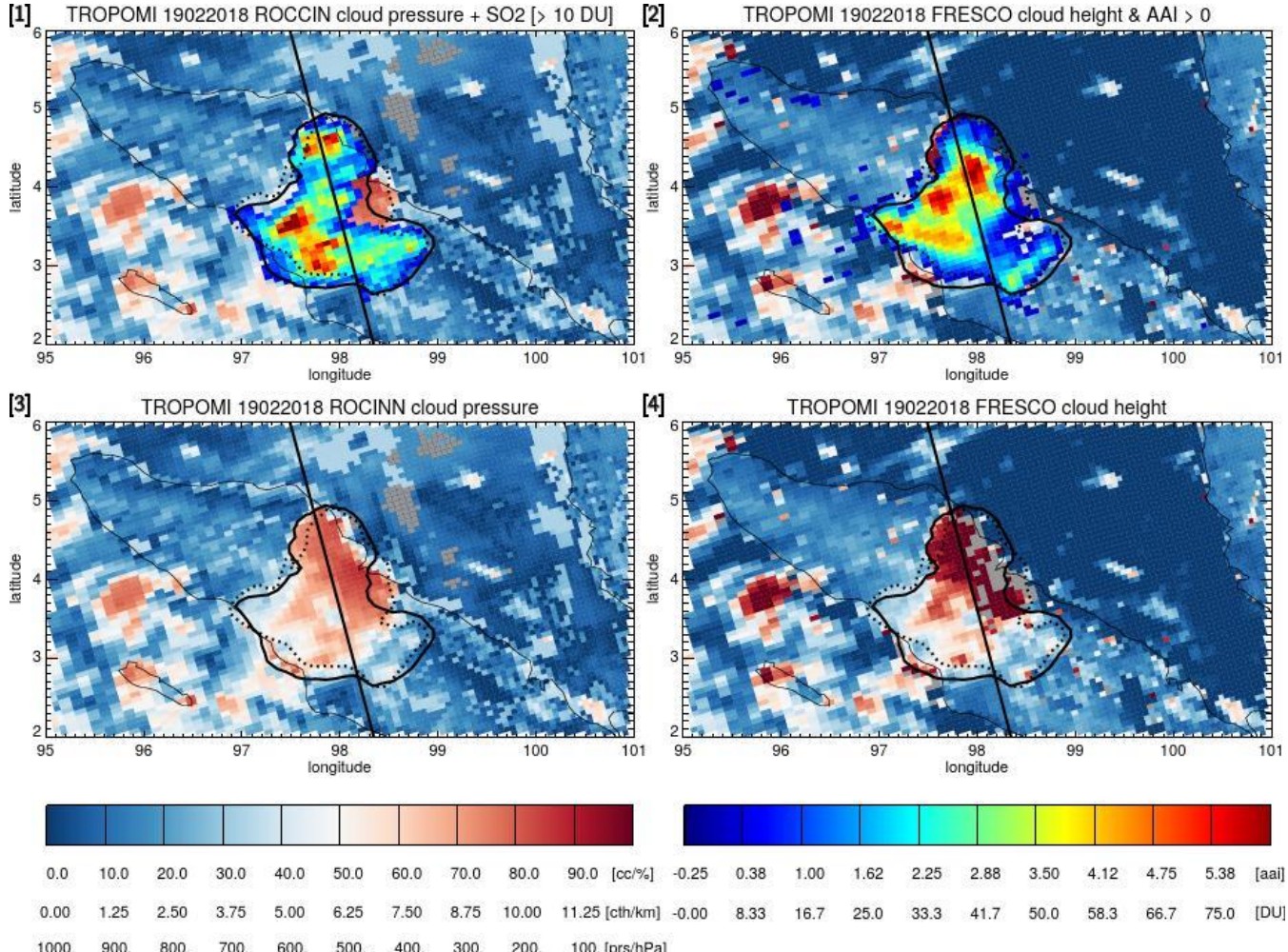

**Figure 1A.** TROPOMI cloud pressure (ROCINN, panels [1] + [3]), and TROPOMI FRESCO cloud heights (panels [2] + [4]). TROPOMI $SO_2$ (panel [1]) and the AAI (panel [2]) for the overpass of the 19 February 2018 Sinabung eruption. The straight

line denotes the path of the CALIPSO overpass, the solid line shape denotes the outline of > 10 DU $SO_2$ columns, the dotted line shape denotes the AAI > 0 value. Note that for FRESCO and ROCINN cloud heights certain pixels are greyed out ("no data"), related to yet unresolved retrieval artefacts.

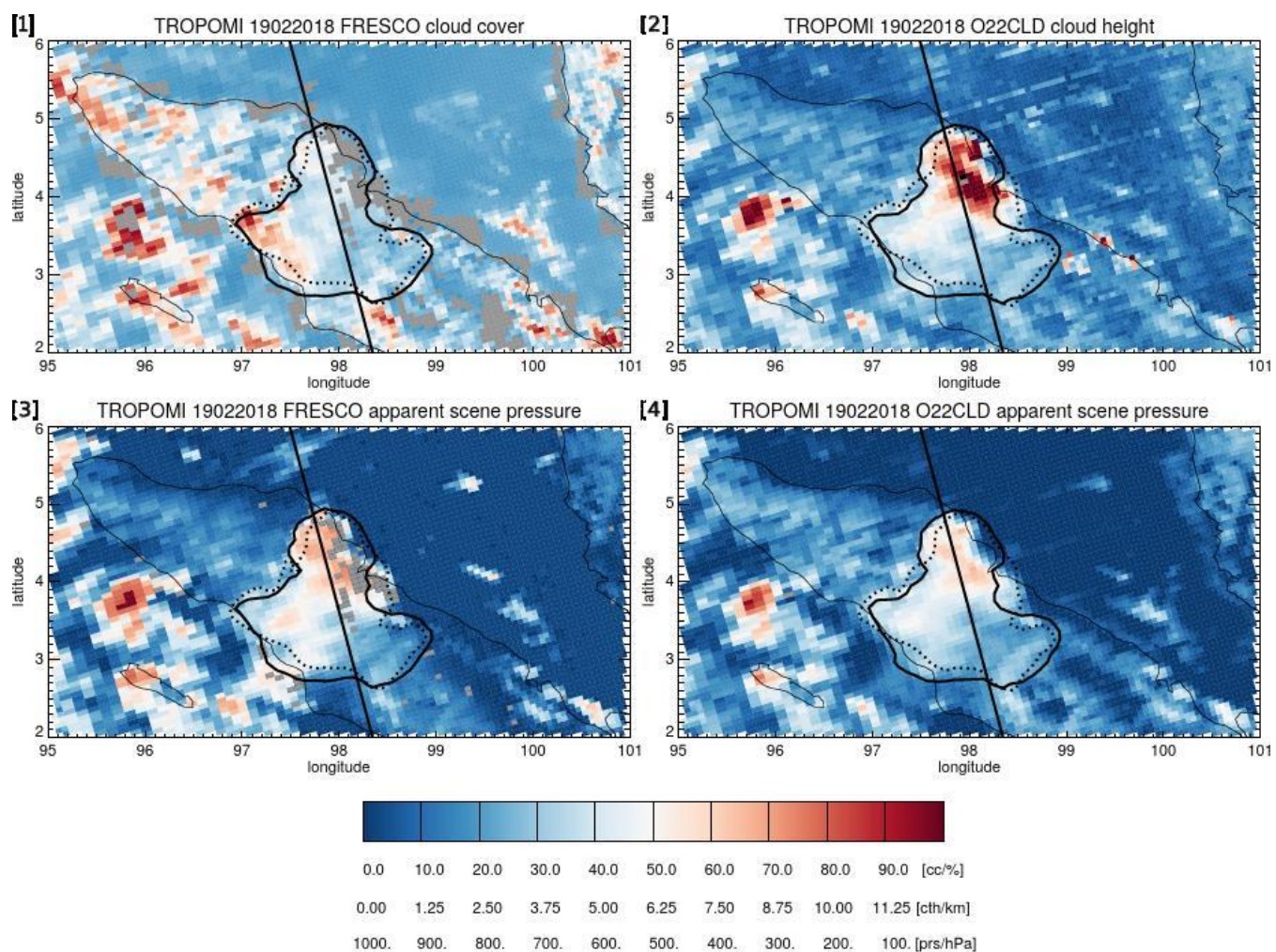

**Figure 1B.** As figure 1A but for TROPOMI FRESCO cloud cover (panel [1]), O22CLD cloud height (panel [2]), FRESCO

apparent scene pressure (panel [3]) and O22CLD apparent scene pressure (panel [4].

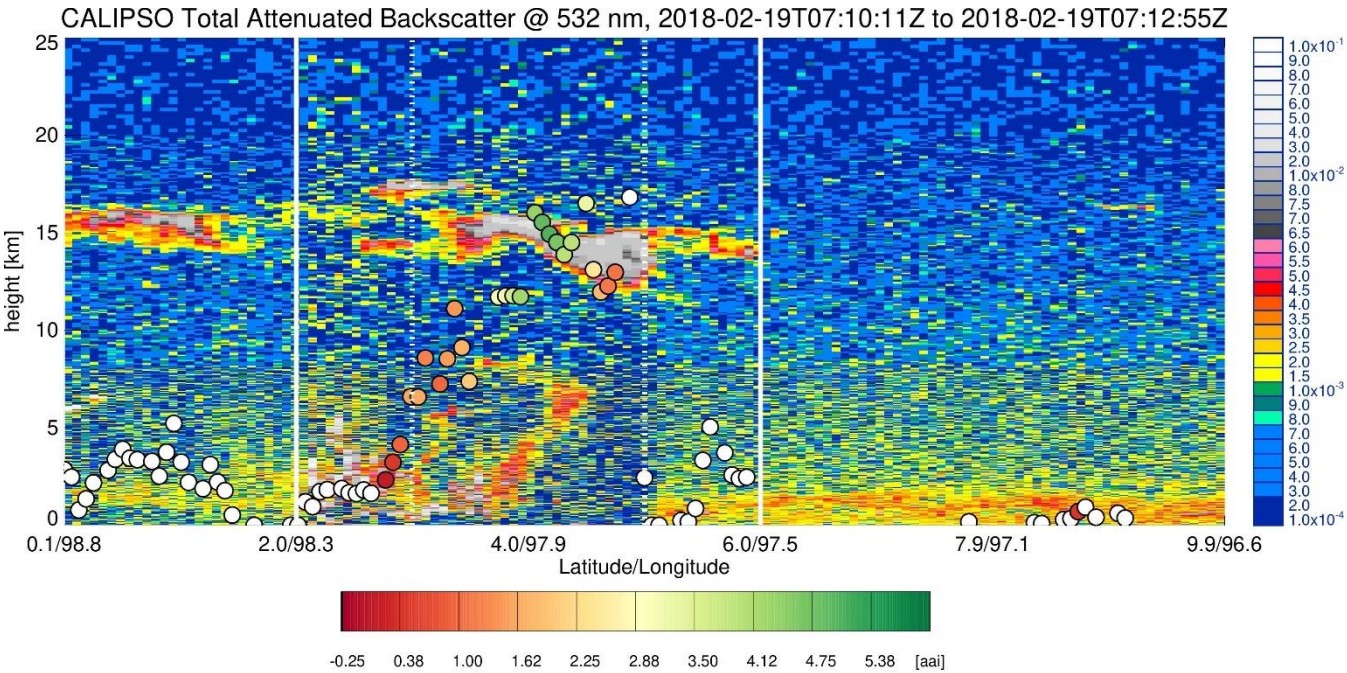

CALIPSO Total Attenuated Backscatter @ 532 nm, 2018-02-19T07:10:11Z to 2018-02-19T07:12:55Z

**Figure 2.** CALIOP total attenuated backscatter profile for the Sinabung eruption on 19 February 2018 along the track indicated in Figure 1. The circles denote the TROPOMI FRESCO cloud heights, color coded according to the TROPOMI AAI values as in figure 1. White dots indicate AAI values < 0.

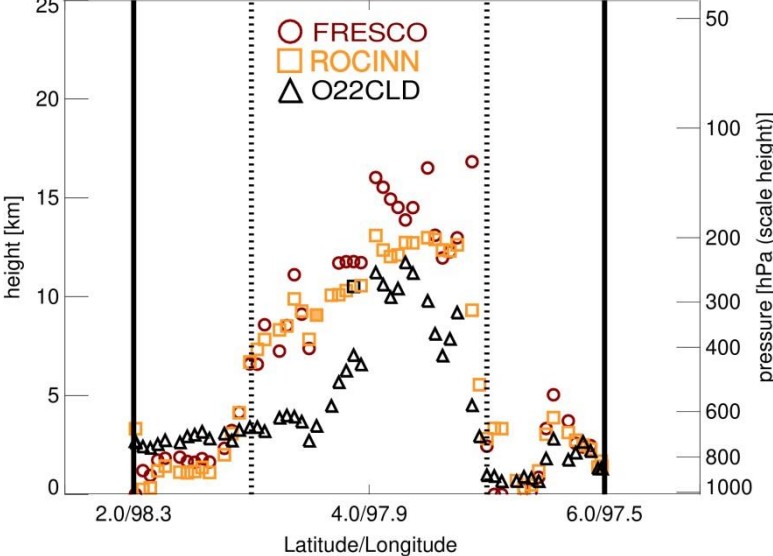

**Figure 3.** TROPOMI cloud heights from the FRESCO, ROCINN and O22CLD algorithms. The solid vertical lines denote the 2°N and 6°N latitudes, the dotted vertical lines the 3° and 5° latitudes. The FRESCO data is identical to the FRESCO data shown figure 2.

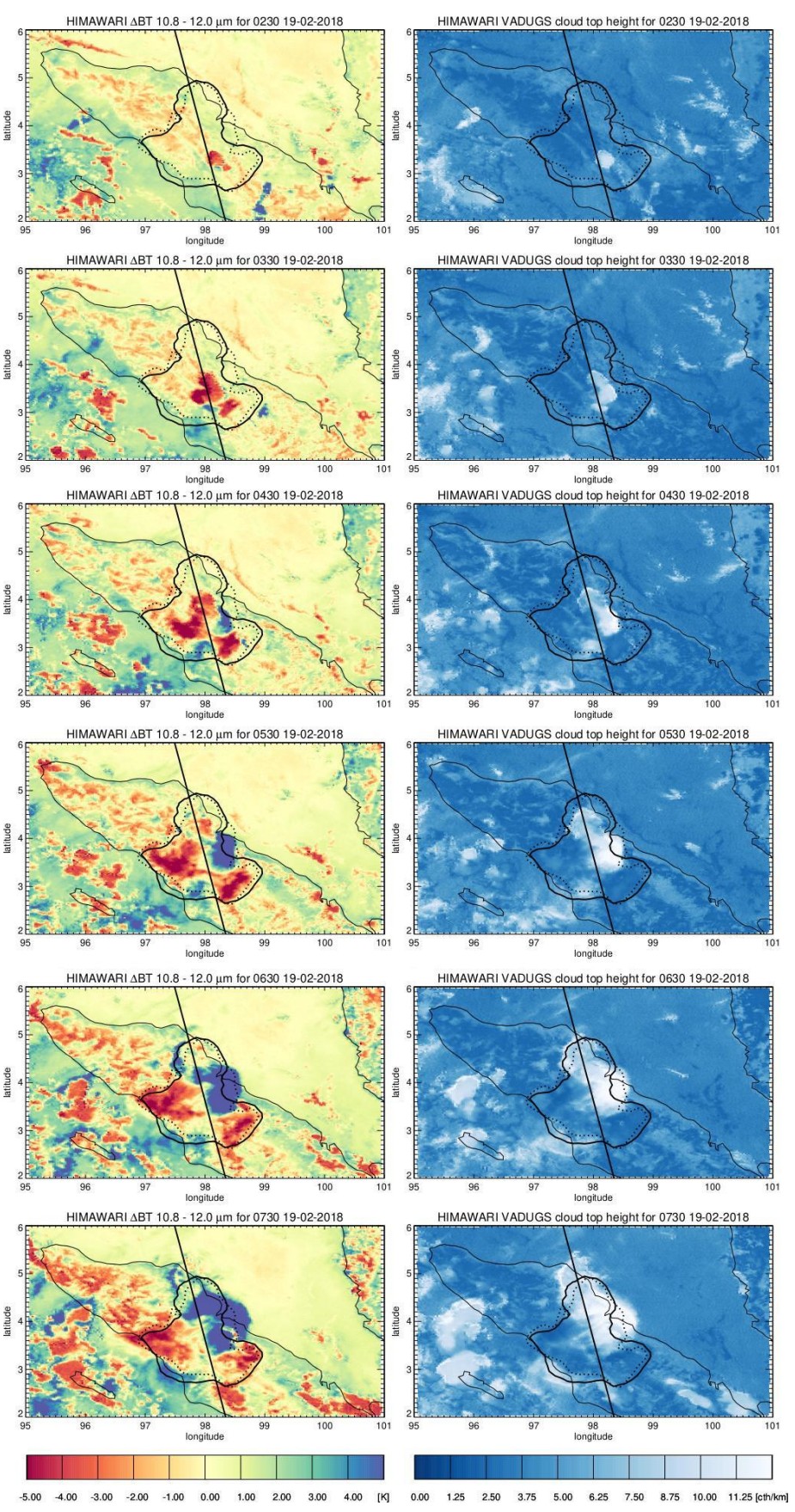

**Figure 4.** Himawari-8 VADUGS cloud heights (right) and 10.8-12.0 µm ΔBTs (left) for every hour between 02:30 and 07:30 UTC. The line denotes the CALIPSO overpass track. The solid and dotted contours denote outline of TROPOMI > 10 DU $SO_2$ columns and TROPOMI AAI > 0 value, as shown in Figure 1.

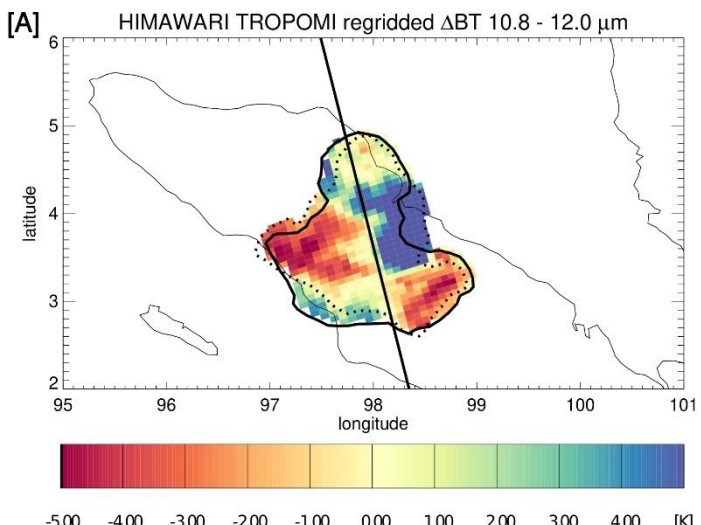

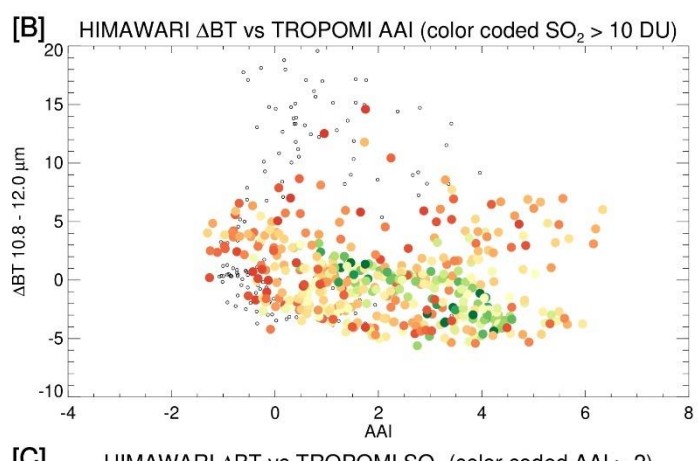

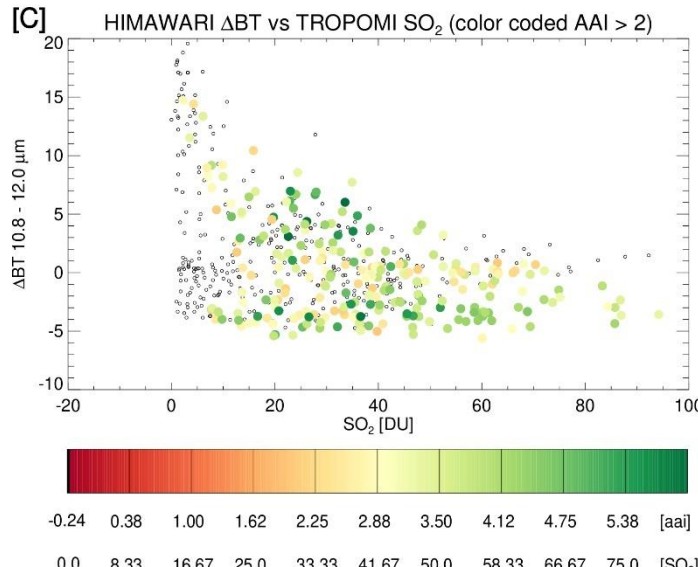

**Figure 5. [A]** Himawari-8 ΔBTs for 19 February 2018 06:30 UTC (see also Figure 4) regridded to the TROPOMI measurement grid of that day, and correlations between the Himawari-8 ΔBTs and TROPOMI **[B]** AAI and **[C]** $SO_2$. The solid and dotted contours denote outline of TROPOMI > 10 DU $SO_2$ columns and TROPOMI AAI > 0 value, as also shown in figure 4 and shown in Figure 1. The color coding of the dots in the AAI scatterplots is indicative of the corresponding $SO_2$ value (> 10 DU) , and the color coding in the $SO_2$ scatterplot is indicative of the AAI value (AAI > 2), see also the lower color bar. These color codings were added for qualitatively identifying possible relationships between ΔBT and AAI or $SO_2$ within the volcanic ash cloud.
