# Peer review of "Analysis of properties of the 19 February 2018 volcanic eruption of Mount Sinabung in S5P/TROPOMI and HIMAWARIHimawari-8 satellite data"

_Natural Hazards and Earth System Sciences, 2019_

## Referee Comment (RC1) · Anonymous Referee #1 · 30 Aug 2019

**General comments**

In this paper, de Laat et al. present a multi-platform, multi-sensor satellite re-mote sensing analysis of an eruption of Mount Sinabung (Indonesia) in 2018. In particular they focus on the 19 February 2018 paroxysmal eruption which produced a volcanic eruption cloud made up of ice-rich, ash-rich and $SO_2$-rich components. The analysis focusses on a validation of three TROPOMI height retrieval algorithms (FRESCO, ROCINN and O22CLD) against a fortuitous CALIPSO lidar (CALIOP) pass that intersected the nascent volcanic cloud. Himawari-8 data is used to interpret the composition, evolution and height of the volcanic cloud and comparisons are made
between the Himawari-8 ∆BT, TROPOMI AAI and SO2 retrievals. The analysis is suitable for NHESS. However, I have some major concerns about the interpretation of the data presented and the relationships drawn between the TROPOMI AAI and Himawari-8 ∆BT values. There are also numerous points of clarification needed throughout the manuscript. These concerns are expressed in the Specific comments and Technical corrections sections below. These comments should be addressed before the manuscript is considered for publication in NHESS.

**Specific comments**

My major concern is with the relationships drawn between the brightness temperature difference (∆BT), Absorbing Aerosol Index (AAI) and SO2 total column amounts (i.e. Fig. 5). By eye, it looks like there is no correlation at all. However, it's difficult to tell as no statistical metrics are given. I suggest adding a statistical metric (perhaps a correlation coefficient if the relationship is expected to be linear) to demonstrate that there's a notable relationship (as the authors claim). Further comment on this is provided in the Technical corrections section.

Another concern is the reliance and interpretation of the 'VADUGS' algorithm. The authors refer to a conference talk, which in general is fine, but as some of the conclusions reached rely on an understanding of this algorithm and its uncertainties a reference to a published article describing it is necessary (in my opinion). If it is not published elsewhere, then a section describing the algorithm should be added if it is to be used in the comparison of the TROPOMI data.

Another issue is the interpretation of the CALIOP data. The CALIOP pass clearly showed a feature that reached 18 km (asl). This is not mentioned anywhere. In addition, the feature (on average) reached cloud-top heights of 16 km (the authors cite a height of 15 km). It is important to get this right as this paper could be a nice

reference for the eruption height of the 19 February 2018 Sinabung cloud in the future.

In general, the authors use the terms ash plume, volcanic cloud, volcanic ash, volcanic ash plume, aerosols etc interchangeably to refer to the eruption cloud (and in some cases to refer to components of the volcanic cloud that were ice-rich). I suggest that the authors define these terms early on in the manuscript. This will avoid confusion, especially when discussing the microphysical make-up of the volcanic cloud. One suggestion could be to use the generic term 'volcanic cloud' to refer to a cloud of volcanic origin and use the terms ice-rich, ash-rich and SO2-rich to refer to regions of the volcanic cloud that exhibit these spectral signatures.

**Technical corrections**

Title: Please be consistent with the use of 'Himawari'. In the text and section headings, the authors use all capital letters in some cases (it is not an acronym). I suggest using 'Himawari-8' throughout the manuscript instead of just 'Himawari' as this is the platform that is used for the analysis (there is a Himawari-9 now, so the distinction is important).

P1L18: 'Evaluation of corresponding Himawari geostationary height retrievals based on InfraRed (IR) brightness temperature differences...' - This statement doesn't seem correct to me. It's the evaluation of the brightness temperature differences (not the height retrieval) that indicates whether the volcanic cloud contains ash or ice/water particles. I suggest changing to 'Evaluation of Himawari-8 geostationary InfraRed (IR) brightness temperature differences...'.

P2L35: 'global monitoring of volcanic eruptions..' - Volcanic Ash Advisory Centers began to do this in 1987 under the International Airways Volcano Watch (see Lechner et al., 2017). This could be mentioned here as well.

P2L57: Change 'like' to 'such as'.

P3L71: Change 'improved heights' to 'improved height retrievals'.

P3L78,L80: Please check the citation style for Smithsonian reports. I don't know which report the authors are referencing. The citation styles used in these two lines are different and I only see one reference to the Smithsonian Institute in the References section. I suggest using their guidelines (i.e. 'Cite this Report' link) for referencing reports.

P3L84: Change 'Calipso' to 'CALIPSO'.

P3L85: '13:30' is this local time (LT) or UTC?

P3L88: Change 'from combining' to 'by combining'.

P5L138: 'column mass load' - Please provide units. Also, to be consistent with the literature, the authors could use 'column mass loading'.

P5L145: 'attenuated backscatter imagery' - Please be more specific. Is this the level 1 version 4, 532 nm total attenuated backscatter product (L1-Standard-V4-10)? There were several recent changes to the CALIPSO lidar calibration from version 3 to 4. Also an up-to-date reference could be added (a series of papers on the new version are published in AMT).

P5L153-155: What about wind shear? This is a common effect known to disperse volcanic ash at different altitudes and different directions. I think it would be worth adding an atmospheric sounding figure to help understand the role of vertical

wind shear.

P5L158: 'local time of 06:25 UTC' - Is this local time or UTC?

P6L163: 'separation between aerosols and SO2' - I would say volcanic ash and SO2. As use of the term 'aerosols' could be interpreted as 'sulphate aerosols'.

P6L164: 'Cook et al. (2014)'. Could add references to Moxnes et al. (2014) and Prata et al. (2017), which both specifically investigate the separation mechanisms of volcanic ash and SO2.

P6L166-168: I would consider moving the VIIRS and NOAA/CIMSS volcanic ash retrieval Supplementary Figures into the main manuscript. The true colour VIIRS image is important for context and interpretation of the TROPOMI height retrievals (presented in Fig. 1). Also, use of NOAA/CIMSS retrievals (which are referred to for the cloud height in this sentence) should be stated with the correct references (i.e. Pavolonis et al. 2015a, b).

P6L171: 'Systematically higher' - This implies FRESCO cloud heights are always higher than the O22CLD heights. Based on Fig. 3, this looks to be the case from 3-5 degrees latitude. However, from 2-3 degrees latitude it looks like O22CLD is higher than FRESCO. So, I wouldn't call this systematic. Perhaps it would be simpler to state 'In general, FRESCO cloud heights are higher than the O22CLD heights'. Or something similar. A correlation plot could also be added to show the bias of FRESCO/ROCINN vs. O22CLD cloud heights.

P6L176: 'up to 15 km altitude' - Please provide a reference for this. Also, how strict is this limit? In Fig. 3, I see cloud heights higher than 15 km. Also, is this above sea level? Please make this clear in the text.

P6L183: Please provide a colour scale/legend with Fig. 2 to show which AAI values correspond to which colour.

P6L183: Interpretation of the CALIOP data. Based on Fig. 2, it looks like the main feature has cloud-top heights of around 16 km (15 km is stated in the manuscript). There is also a clear feature at 18 km (detected by the AAI). This is not mentioned at all and should be addressed in the manuscript.

P6L188-189: There is poor agreement between FRESCO and CALIOP from 3-4 degrees latitude, which should be stated here.

P7L193: CALIOP's feature mask - Please state which version of the feature mask is being interpreted. There were changes made from V3 to V4. I looked at the VFM V4 for this pass and I can see some small parts of the feature classified as dust aerosol but the majority is cloud.

P7L194: 'clearly the attenuation is not complete.' - I'm not sure it is that clear. This interpretation would be more justified if the VFM was plotted on the same scale as Fig. 2 and inserted as a second panel.

P7196: Comparison of FRESCO and ROCINN - is this only for AAI $> 0$? Please clarify.

P7L200-201: 'and all data products increasing heights in the volcanic cloud going from south to north.' - This is simply not true. The heights increase from 2-4 degrees latitude and then decrease from 4-6 degrees latitude. Please clarify in the text.

P7L209-210: 'The eruption dynamics may thus have additional effects on the ash plume displacement, but this cannot be investigated based on the available

satellite data.' - This statement requires further justification and clarification about why the available satellite data cannot be used to study the eruption dynamics. For example, Himawari-8 provides excellent observations (every 10 minutes) of the volcanic cloud's evolution and dynamics (as the authors discuss in Section 3.4).

P7L212: Change 'is' to 'was'.

P7L216: Positive $\Delta$BTs are also indicative of clouds composed of water droplets (not just ice). Please clarify in the text.

P7L218-219: 'one associated also with a high cloud height, and another one further south with much lower cloud heights' - what cloud heights are being referred to here? The VADUGS algorithm in Fig. 4 only appears to show a high altitude cloud.

P7L221: 'dense high ice clouds' - What do the authors mean by 'dense' here? Optically thick ice clouds would show a near zero $\Delta$BT, not a strongly positive $\Delta$BT.

P7L221: 'purple region' - I actually see this as blue. Maybe call it an 'ice-rich cloud'?

P7L222-224:
Figure 5 - I found this figure difficult to interpret. At this line in the manuscript the authors refer to the 'HIMAWARI VADUGS $\Delta$BTs'. How are VADUGS $\Delta$BTs different to a simple 11-12 micron $\Delta$BT? In Fig. 5 they just look like $\Delta$BTs. The authors also state that 'When focusing on AAI and SO2 values, it appears that larger $\Delta$BT values occur for smaller AAI values ($<$ 2) and SO2 ($<$ 20 DU)' - For the lower left plot in Fig. 5, I can see numerous data points that have positive $\Delta$BT (0-10 K) for large (2-6) positive AAI values (contradictory to what the authors claim) and I find it very hard to interpret any relationship whatsoever in this panel. In Fig. 5 lower right panel, again, it's hard to see

any relationship because there are positive and negative ΔBT values that correspond to a whole range of SO2 values (5-100 DU).

There are several ways Fig. 5 could be improved:
First, I would only plot the data that falls within the contours plotted in the upper left panel of Fig. 5 as this clearly contains the volcanic cloud (what are these contours by the way? They are not mentioned in the Fig. 5 caption). This would remove the black dots (I assume?), which at the moment are distracting. Second, some kind of statistical metric could be used to indicate that there is indeed a relationship between AAI, ΔBT and SO2. If the relationship is not linear then maybe some kind of curve fit (exponential for lower right panel?) will help the reader interpret the relationships.

P7L224: 'The larger ΔBT are also associated with optically more dense clouds (see VIIRS imagery in the SI and comparison of TROPOMI with CALIPSO).' - This statement needs to be further clarified. It's not physically possible for an optically thick cloud to have a large ΔBT in the infrared. When clouds become optically thick they behave as grey bodies (little spectral variation across thermal infrared wavelengths) and so a difference in brightness temperature between 11 and 12 micron should be close to zero. However, I think what the authors are observing is a relationship between high reflectance at visible wavelengths (white clouds in VIIRS imagery) and large ΔBTs, but it's not clear in the way that it's stated.

P8L225-227: This could be due to scavenging of SO2 by ice (Rose et al., 2000). It could also be due to ice nucleation of volcanic ash particles (Durant et al., 2008). In terms of the conversion of SO2 to sulphate, is there a reference that could be added here? i.e. how long does it typically take for SO2 to convert to sulphate in the upper troposphere? And does this conversion rate make sense given the time of observation and time since eruption?

P8L245-248: 'Comparison with geostationary IR volcanic ash height' - Which re-trieval is this statement referring to? Is this the VADUGS volcanic ash cloud height retrieval? It's the comparison with CALIOP that demonstrates TROPOMI height algorithms may underestimate heights for semi-transparent ash clouds. Please clarify this.

P8L251-252: The 'shielding' effect - This is rather speculative and could be due to a number of different reasons (see previous comments on P8L225-227). Also, is this shielding of SO2 or ash or both? I think to substantiate this claim, evidence of SO2/ash existing underneath the cloud-top should be provided.

P9L257-258: 'the retrieval algorithm' - which retrieval algorithm is being referred to here? Please clarify.

P9L266-268: 'TROPOMI cloud heights can be used for determining aerosol heights for AAI values greater than 4' - How was this conclusion reached? What is the significance of AAI $> 4$. As stated in the previous sentence, the TROPOMI cloud heights do not perform well for semi-transparent clouds regardless of their AAI value. This statement requires further clarification. Also 'column values $> 1$ DU' is TROPOMI's signal-to-noise really this good? Please provide a reference.

P11L323-325: Please fix reference formatting here. Also link provided to Stein-Zweers (2016) results in a 'Page not found' error.

P16L411: Check style for figure labels e.g. 'A+E' should be '(a) and (e)'.

P19L415: VADUGS cloud heights are on the right column of Fig. 4 not left and the $\triangle$BTs are on the left.

P19L427: Change 'derived' to 'shown'.

P19L427: What is the $\Delta$BT bias correction? This needs to be explained and defined in the manuscript.

**References**

Durant, A. J., Shaw, R. A., Rose, W. I., Mi, Y. and Ernst, G. G. J.: Ice nucleation and overseeding of ice in volcanic clouds, Journal of Geophysical Research, 113(D9), doi:10.1029/2007JD009064, 2008.

Lechner P, Tupper A, Guffanti M, Loughlin S, Casadevall T. Volcanic Ash and Aviation—The Challenges of Real-Time, Global Communication of a Natural Hazard. In Observing the Volcano World, 2017 (pp. 51-64). Springer, Cham.

Moxnes, E. D., Kristiansen, N. I., Stohl, A., Clarisse, L., Durant, A., Weber, K. and Vogel, A.: Separation of ash and sulfur dioxide during the 2011 Grímsvötn eruption, Journal of Geophysical Research: Atmospheres, 119(12), 7477–7501, doi:10.1002/2013JD021129, 2014.

Pavolonis, M. J., Sieglaff, J. and Cintineo, J.: Spectrally Enhanced Cloud Objects-A generalized framework for automated detection of volcanic ash and dust clouds using passive satellite measurements: 1. Multispectral analysis: ASH/DUST DETECTION, PART 1, Journal of Geophysical Research: Atmospheres, 120(15), 7813–7841, doi:10.1002/2014JD022968, 2015a.

Pavolonis, M. J., Sieglaff, J. and Cintineo, J.: Spectrally Enhanced Cloud Objects-A generalized framework for automated detection of volcanic ash and dust clouds using passive satellite measurements: 2. Cloud object analysis and global application:

ASH/DUST DETECTION, PART 2, Journal of Geophysical Research: Atmospheres, 120(15), 7842–7870, doi:10.1002/2014JD022969, 2015b.

Prata, F., Woodhouse, M., Huppert, H. E., Prata, A., Thordarson, T. and Carn, S.: Atmospheric processes affecting the separation of volcanic ash and SO2 in volcanic eruptions: inferences from the May 2011 Grímsvötn eruption, Atmospheric Chemistry and Physics, 17(17), 10709–10732, doi:10.5194/acp-17-10709-2017, 2017.

Rose, W. I., Bluth, G. J. S. and Ernst, G. G. J.: Integrating retrievals of volcanic cloud characteristics from satellite remote sensors: a summary, edited by P. Francis, J. Neuberg, and R. S. J. Sparks, Philosophical Transactions of the Royal Society of London. Series A: Mathematical, Physical and Engineering Sciences, 358(1770), 1585–1606, doi:10.1098/rsta.2000.0605, 2000.
* * *

---

## Referee Comment (RC2) · Anonymous Referee #2 · 5 Nov 2019

In the paper "Analysis of properties of the 19 February 2018 volcanic eruption of Mount Sinabung in S5P/TROPOMI and Himawari satellitedata" de Laat et al present a multi-sensor approach to determine the altitude of the volcanic ash cloud from the February 2019 Sinabung eruption. The results from different cloud retrieval algorithms applied to TROPOMi data are compared to volcanic ash heights based on CALIPSO data as well as from Himawari geostationary satellite data. Although the analysis is suitable for NHESS, some major changes to the text and figures are request, as described in the specific comments below.

Specific comments:

[Figure]

P3, L66: Missing word '(down to 2.5x7km2) of SO2'

P3, L70. The term "volcanic clouds" is mis-leading, since it can refer to clouds of ash, particles, trace-gases. Here you are clearly referring to volcanic ash clouds. You use this term very often throughout the text. Please describe at each occasion what you mean.

P3, L84: Calipso -> CALIPSO

P3, L85. Maybe add a short explanation about what the 'A-train constellation' is.

P3, L85. It is "13:30h local time"

P4, L118: Add here that the SO2 product provides four different SO2 VCDs for different SO2 vertical profile shapes, since they are not known at the time of the measurement. For the rest of the paper it would be also good to know, which SO2 VCD you have chosen. Here you might also refer to the paper of Hedelt et al. 2019, who has also studied the Sinabung eruption and retrieved SO2 plumeheights for this.

P4, L123: Add a reference for the O22CLD algorithm (either paper or ATBD)

P4, L129. Consider also adding information about the cloud fraction from OCRA

P5, Sect. 2.5: I suggest to add more information on CALIOP, references and a description of what the 'attenuated backscatter imagery' displays, i.e. what it is sensitive to, etc. I also propose to also add the VFM, which shows the type of absorption feature as well as the BTD which gives information about the type of absorption.

P6, L163 You write the 'extend of the volcanic plume', but by means of what? SO2 VCD or AAI or? Please specify.

P6, L166: Here it would be interesting to see what is the TROPOMI OCRA cloud fraction.

P6, L170 Please describe the 'clear differences' between FRESCO and ROCINN

[Figure]

P6, L183 I suggest to rephrase the sentence, since the CALIOP data only shows an attenuation by clouds. As you write later on, there is no *CLEAR* detection of an ash layer

P6, L187: Add the CALIPSO overpass time here, such that the reader gets an idea about the overpass time difference btw TROPOMI & CALIPSO

P7, L193. The VFM classifies the volcanic cloud as 'cloud' and sometimes 'ash'. This is because fresh volcanic plumes are typically rich in water vapor (especially for tropical eruptions). The volcanic clouds also contain high concentrations of waterdroplets. Therefore, the classification in the CALIPSO VFM sometimes fails to pick up the volcanic ash or sulfate aerosol because of competing clouds. Another interesting feature which could be analyzed in this paper is the brightness temperature difference from CALIPSO which clearly shows the ash in the data

P7 L214-216. The description of the BTD should appear in Sect. 2.4

P8 L237: TROPOMI was launched in 2017. Given that we now have 2019, I wouldn't call it 'recently launched'.

P8 L225-226 Since the ash and SO2 cloud are co-located there is certainly also an effect of the ash on the SO2 VCD retrieval (and not only shielding). Might be interesting to study the effect of ash on the SO2 VCD

Figures

Figure 1 is clearly overloaded. Although it is interesting to see all results in one single figure, it is really hard to understand all plots. I would suggest to break down the figure into several figures (i.e. show FRESCO/OCRA/O22CLD cloud heights separately in a figure, as well as FRESCO/OCRA CF and SO2/ AAI) before showing combined plots

Figure 2: Clearly a colorbar for the attenuation backscatter is missing. Also please add the AAI colorbar to this figure and not only refer to it. Please consider to show also the VFM, showing some ash classification in the cloud as well

Figure 5: I would rearrange the plots and show the HIMAWARI vs AAI plots next to each other and clearly indicate the SO2 threshold in the plot title. A color bar for each subplot would also be very helpful. Have you tried to display these results in a 3D scatter plot, with x=SO2, y=AAI and z=BTD? Furthermore, why did you show the scatterplots for AAI < -0.25 – they are not part of the volcanic ash cloud, as your AAI contours also indicate. For the SO2 plot I suggest using a logarithmic scale. I also suggest showing horizontal/vertical lines at x=0 and y=0

Figures S1, S2a, S2b: Would it be possible to indicate the location of the volcano on the maps?

Figures S2: Would it be possible to add the AAI/SO2 contours to the maps?

---

## Author Comment (AC1) · 6 Feb 2020

**Response to referee #1**

Specific comments

My major concern is with the relationships drawn between the brightness temperature difference (ΔBT), Absorbing Aerosol Index (AAI) and SO2 total column amounts (i.e. Fig. 5). By eye, it looks like there is no correlation at all. However, it's difficult to tell as no statistical metrics are given. I suggest adding a statistical metric (perhaps a correlation coefficient if the relationship is expected to be linear) to demonstrate that there's a notable relationship (as the authors claim). Further comment on this is provided in the Technical corrections section.

- *there is indeed no clear relation between ΔBT, AAI, and SO2, which is an important aspect of the paper. If anything, our results suggest that the relation is complex because the observations were made shortly after the eruption, and the presence of ash and condensed water. We find indications that the presence of ash and condensed water causes a possible shielding effect, i.e. TROPOMI only observes some aspects of the volcanic plume, while the presence of condensed water also hampers the IR ΔBT detection of volcanic ash (a known problem, but nevertheless problematic if these type of measurements are to be used in for example aviation applications).*

   *with the help of the referee comments and changes we made we think this should be clearer, see further in the responses to the referee comments.*

Another concern is the reliance and interpretation of the 'VADUGS' algorithm. The authors refer to a conference talk, which in general is fine, but as some of the conclusions reached rely on an understanding of this algorithm and its uncertainties a reference to a published article describing it is necessary (in my opinion). If it is not published elsewhere, then a section describing the algorithm should be added if it is to be used in the comparison of the TROPOMI data.

- *This is a sensitive and difficult point. We currently do not have the capacity nor the time to extensively describe VADUGS, which in essence would be a DLR task, as VADUGS is their algorithm.*

   *Originally we expected that VADUGS – which has been an operational algorithm at DLR since 2013 – would be described in a separate paper independent of the EUNADICS-AV project by DLR and by the time our paper was submitted. Due to several staffing changes at DLR (including the DLR co-author moving to EUMETSAT) this has not yet happened.*

   *However, we would argue that the paper actually does not rely that much on VADUGS but rather mostly on HIMAWARI-8 IR ΔBT differences, which is identical for any volcanic ash retrieval algorithm. We do use he retrieved VADUGS heights (but not ash amounts, not effective radius), but only in a qualitive sense.*

   *As the reliance on specific and quantitative VADUGS results is limited, the lack of official VADUGS description should not have to be considered as overly important.*

   *However, so this is to decide by the editor. And we reiterate that unfortunately there is little we can do to resolve this point by providing a full reference to the VADUGS algorithm.*

Another issue is the interpretation of the CALIOP data. The CALIOP pass clearly showed a feature that reached 18 km (asl). This is not mentioned anywhere. In addition, the feature (on average)

reached cloud-top heights of 16 km (the authors cite a height of 15 km). It is important to get this right as this paper could be a nice reference for the eruption height of the 19 February 2018 Sinabung cloud in the future.

- *we have added the observation that there also appears to be (some) ash up to 18 km, and explained why this layer likely is also of volcanic origin as there are no indications from HIMAWARI-8 that there are other high altitude clouds at those locations at that time. This does not impact the findings of the analysis, as this layer coincides with the thinner part of the volcanic plume for which we argued the TROPOMI height retrievals are less accurate as TROPOMI "sees" partly through the cloud.*

In general, the authors use the terms ash plume, volcanic cloud, volcanic ash, volcanic ash plume, aerosols etc interchangeably to refer to the eruption cloud (and in some cases to refer to components of the volcanic cloud that were ice-rich). I suggest that the authors define these terms early on in the manuscript. This will avoid confusion, especially when discussing the microphysical make-up of the volcanic cloud. One suggestion could be to use the generic term 'volcanic cloud' to refer to a cloud of volcanic origin and use the terms ice-rich, ash-rich and SO2-rich to refer to regions of the volcanic cloud that exhibit these spectral signatures.

- *we have changed all references to "volcanic ash plume" into "volcanic ash cloud", all references to "volcanic ash height" into "volcanic ash cloud height". This leaves essentially two expressions: "volcanic ash cloud (height)", which refers to the volcanic eruption cloud phenomon, and "volcanic ash", which refers to the physical quantity. We checked the document to ensure that "volcanic ash" is only used when we refer to volcanic ash as the physical quantity.*

**Technical corrections**

Title: Please be consistent with the use of 'Himawari'. In the text and section headings, the authors use all capital letters in some cases (it is not an acronym). I suggest using 'Himawari-8' throughout the manuscript instead of just 'Himawari' as this is the platform that is used for the analysis (there is a Himawari-9 now, so the distinction is important).

- *All references to HIMAWARI have been replaced with HIMAWARI-8 (incl. figure captions)*

P1L18: 'Evaluation of corresponding Himawari geostationary height retrievals based on InfraRed (IR) brightness temperature differences...' - This statement doesn't seem correct to me. It's the evaluation of the brightness temperature differences (not the height retrieval) that indicates whether the volcanic cloud contains ash or ice/water particles. I suggest changing to 'Evaluation of Himawari-8 geostationary InfraRed (IR) brightness temperature differences...'.

- *Changed lines 19-21 to:*

  *"Evaluation of corresponding HIMAWARI-8 geostationary InfraRed (IR) brightness temperature differences (ΔBT) - a signature for detection of volcanic ash in geostationary satellite data and widely used as input for quantitative volcanic ash retrievals - reveals that for this particular eruption the ΔBT volcanic ash signature changes to a ΔBT ice crystal signature for the part …"*

P3L78,L80: Please check the citation style for Smithsonian reports. I don't know which report the authors are referencing. The citation styles used in these two lines are different and I only see one reference to the Smithsonian Institute in the References section. I suggest using their guidelines (i.e. 'Cite this Report' link) for referencing reports.

- *Deleted the reference in the reference list, changed the "in-text" reference to:*

  "[https://volcano.si.edu/volcano.cfm?vn=261080; Eruptive History]"

P3L85: '13:30' is this local time (LT) or UTC?

- *changed to*

  *"The TROPOMI equator crossing local time of 13:30"*

P5L145: 'attenuated backscatter imagery' - Please be more specific. Is this the level 1 version 4, 532 nm total attenuated backscatter product (L1-Standard-V4-10)? There were several recent changes to the CALIPSO lidar calibration from version 3 to 4. Also an up-to-date reference could be added (a series of papers on the new version are published in AMT).

- *Results presented are CALIPSO v3.40 data. For the qualitative use of the TAB data product (actual TAB values) in this study and the purpose of this study, differences in the TAB between both product versions are marginal/negligible. We found it very hard to find differences in a comparison of figure 1 between both v3.40 and v4.10. The choice for v3.40 data was pragmatic as data files at the time for v3.40 were earlier available than data files for v4.10*

- *changed sentence to:*

" … we use total attenuated backscatter data from one CALIPSO orbit (data version 3.40) in a qualitative approach, *i.e.* detection of cloud and aerosol layers and their heights."

P5L158: 'local time of 06:25 UTC' - Is this local time or UTC?

- *'local' here refers to the geographical location, as measurement times change during a TROPOMI orbit. This is indeed confusing, so we changed the sentence to:*

   "with TROPOMI measurements within the figure area made at approximately 06:25 UTC"

P6L164: 'Cook et al. (2014)'. Could add references to Moxnes et al. (2014) and Prata et al. (2017), which both specifically investigate the separation mechanisms of volcanic ash and SO2

- *references added*

P6L166-168: I would consider moving the VIIRS and NOAA/CIMSS volcanic ash retrieval Supplementary Figures into the main manuscript. The true colour VIIRS image is important for context and interpretation of the TROPOMI height retrievals (presented in Fig. 1). Also, use of NOAA/CIMSS retrievals (which are referred to for the cloud height in this sentence) should be stated with the correct references (i.e. Pavolonis et al. 2015a, b)

- *this is a point of contention: originally the VIIRS and NOAA/CIMSS retrievals actually were in the main paper. However, as (1) the paper is already rich in figures and (2) the main topic of the paper is evaluating TROPOMI and HIMAWARI-8 volcanic ash heights, we decided at the finalizing stage of writing to move the VIIRS/NOAA/CIMSS results to the SI.*

   *Furthermore, in essence the VIIRS/NOAA/CIMSS results are based on the same type of measurements (broad band IR) and the same retrieval approaches (split-channel) as the HIMAWARI- data. However, data files of the VIIRS/NOAA/CIMSS results are not publicly available and accessible, which would allow for direct comparison. Hence, use of results that are only available as imagery, not as data, and we prefer not to rely just on imagery in the main body of the paper.*

   *Unless it is really crucial for the paper – which believe it is not - we would prefer to keep the VIIRS and NOAA/CIMSS figures in the SI.*

- *references have been adjusted as suggested*

P6L171: 'Systematically higher' - This implies FRESCO cloud heights are always higher than the O22CLD heights. Based on Fig. 3, this looks to be the case from 3-5 degrees latitude. However, from 2-3 degrees latitude it looks like O22CLD is higher than FRESCO. So, I wouldn't call this systematic. Perhaps it would be simpler to state 'In general, FRESCO cloud heights are higher than the O22CLD heights'. Or something similar. A correlation plot could also be added to show the bias of FRESCO/ROCINN vs. O22CLD cloud heights.

- *changed sentence as suggested*

- *For general interest: we did a check: for the region of the plot, and for 88% of clouds with CTH > 5 km, FRESCO CTH are higher than O2O2 CTH. For a wider geographical area (entire orbit), this drops to 60%. However, when instead considering clouds with CTH > 10 km for the*

*wider area, this fraction increases again, even further for larger cloud fractions (0-25-50-75% cloud fraction 67-71-76-87%). For ROCINN, these numbers are even better (89-91-93-97%).*

P6L176: 'up to 15 km altitude' - Please provide a reference for this. Also, how strict is this limit? In Fig. 3, I see cloud heights higher than 15 km. Also, is this above sea level? Please make this clear in the text.

- *changed to:*

  "up to approximately 17 km altitude (~100 hPa) [Wang et al., 2012]"

- *This is a typo, it should read "25 km", which is a hard-coded limit for FRESCO CTH. However, a more realistic limit – and hence why we overlooked it, is that in reality FRESCO has – by virtue of its oxygen absorption – an acceptable sensitivity up to approximately 100 hPa or approximately 17 km altitude. Research papers on both FRESCO validation as well as FRESCO methodological uncertainties indicate suggest an uncertainty range of 25-50 hPa. Furthermore, there is limited information about the quality of FRESCO CTH above 15 km there are few validation opportunities. Clouds with clouds top heights at 15 km or above are rare, even more so optically thick clouds for which the FRESCO "centroid altitude" lies at 15 km or above. However, with increased possibilities for measuring clouds above 15 km altitude (more & better satellites), as well as interest in high altitude clouds above 15 km altitude, this will be a valuable future research topic.*

P6L183: Please provide a colour scale/legend with Fig. 2 to show which AAI values correspond to which colour.

- *a color scale was added*

P6L183: Interpretation of the CALIOP data. Based on Fig. 2, it looks like the main feature has cloud-top heights of around 16 km (15 km is stated in the manuscript). There is also a clear feature at 18 km (detected by the AAI). This is not mentioned at all and should be addressed in the manuscript.

- *changed to "approximately 16 km"*

- *added a short discussion about this 18 km plume later in section 3.4 in relation to Figure 4.*

  "There is also a layer detected in CALIPSO at 18 km around 3°N, which likely is also volcanic as the HIMAWARI-8 BT does not provide any indication of other high clouds while there are negative ΔBTs near the CALIPSO track at 3°N, indicative of the presence of volcanic ash."

P6L188-189: There is poor agreement between FRESCO and CALIOP from 3-4 degrees latitude, which should be stated here.

- *changed to:*

  "Between 3° and 4° latitude, the agreement is poor as the FRESCO"

P7L193: CALIOP's feature mask - Please state which version of the feature mask is being interpreted. There were changes made from V3 to V4. I looked at the VFM V4 for this pass and I can see some small parts of the feature classified as dust aerosol but the majority is cloud.

- *added, it is v3.4*

P7L194: 'clearly the attenuation is not complete.' - I'm not sure it is that clear. This interpretation would be more justified if the VFM was plotted on the same scale as Fig. 2 and inserted as a second panel.

- *correct, although even for CALIPSO v4.10 the nearly all cloud pixels for this scene are flagged as "cloud". We changed the sentence to:*

  *"does not identify hardly any of these backscatter signals as aerosol (for CALIOP v4.10 an occasional cloud pixel is flagged as aerosol):"*

P7L196: Comparison of FRESCO and ROCINN – is this only for AAI > 0? Please clarify.

- *The R2 = 0.98 is based on all clouds regardless of AAI value. Sentence changed to:*

  *"FRESCO cloud heights between 0.5 and 14 km regardless of corresponding AAI value"*

P7L200-201: 'and all data products increasing heights in the volcanic cloud going from south to north.' - This is simply not true. The heights increase from 2-4 degrees latitude and then decrease from 4-6 degrees latitude. Please clarify in the text.

- *This is sloppy formulation from our side, the point we wanted to make – and the change we made - is:*

  *"… with the largest heights between 4° and 5° latitude, consistent with the CALIOP observation that backscatter signals between 3° and 4° latitude are weaker than between 4° and 5° latitude"*

P7L209-210: 'The eruption dynamics may thus have additional effects on the ash plume displacement, but this cannot be investigated based on the available satellite data.' - This statement requires further justification and clarification about why the available satellite data cannot be used to study the eruption dynamics. For example, Himawari-8 provides excellent observations (every 10 minutes) of the volcanic cloud's evolution and dynamics (as the authors discuss in Section 3.4).

- *What would be needed for properly understanding the eruption dynamics are time series of its 3D structure (time-lon-lat-height), like in a model, but which clearly cannot be observed or reconstructed from satellite observations. Satellites thus only provide a partial view of the entire eruption plume.*

  *Changed the description to:*

  *"The eruption dynamics may thus have additional effects on the ash plume displacement, for which time series of the complete 3-dimensional view of the eruption plume would be preferred.  The current available satellite data only provide a 2-dimensional view of the eruption plume from above (geostationary, Polar orbiting), with information about changes over time in case of the geostationary satellites and with some but limited information about cloud and aerosol height.  CALIOP measurements only provide one 2-dimensional cross-section through the eruption plume, without any information about changes over time."*

P7L212: Change 'is' to 'was'.

- *changed*

P7L216: Positive ΔBTs are also indicative of clouds composed of water droplets (not just ice). Please clarify in the text.

- *changed to:*

  "with negative ΔBT potentially indicating volcanic ash, and positive ΔBTs indicative of the presence of nontrivial liquid water of ice content [Pavolonis et al., 2006]."

P7L218-219: 'one associated also with a high cloud height, and another one further south with much lower cloud heights' - what cloud heights are being referred to here? The VADUGS algorithm in Fig. 4 only appears to show a high altitude cloud.

- *some reddish colors within the contours, potentially indicating volcanic ash, collocated with high clouds (whites), while other reddish colors collocated with low clouds (blues). To describe exactly what we note, we changes the sentence to:*

  "one associated also with a high cloud height (white cloud colors), and another one further south with much lower cloud heights, likely low-altitude outflow or pyroclastic flows (blue cloud colors)."

P7L221: 'dense high ice clouds' - What do the authors mean by 'dense' here? Optically thick ice clouds would show a near zero ΔBT, not a strongly positive ΔBT.

- *dense refers to the observations that initially the high cirrus is not transparent (see also SI figures S 2A/2B). It is not critical, so we modified the text in combination with the next comment.*

P7L221: 'purple region' - I actually see this as blue. Maybe call it an 'ice-rich cloud'?

- *see also previous comment, sentences were changed, the color purple now only refers to the ΔBTs:*

  "large positive ΔBTs (purple), indicative of high ice clouds, which continues to grow and expand northward."

P7L222-224: Figure 5 - I found this figure difficult to interpret. At this line in the manuscript the authors refer to the 'HIMAWARI VADUGS ΔBTs'. How are VADUGS ΔBTs different to a simple 11-12 micron ΔBT? In Fig. 5 they just look like ΔBTs. The authors also state that 'When focusing on AAI and SO2 values, it appears that larger ΔBT values occur for smaller AAI values (< 2) and SO2 (< 20 DU)' - For the lower left plot in Fig. 5, I can see numerous data points that have positive ΔBT (0-10 K) for large (2-6) positive AAI values (contradictory to what the authors claim) and I find it very hard to interpret any relationship whatsoever in this panel. In Fig. 5 lower right panel, again, it's hard to see any relationship because there are positive and negative ΔBT values that correspond to a whole range of SO2 values (5-100 DU).
There are several ways Fig. 5 could be improved: First, I would only plot the data that falls within the contours plotted in the upper left panel of Fig. 5 as this clearly contains the volcanic cloud (what are these contours by the way? They are not mentioned in the Fig. 5 caption). This would remove the black dots (I assume?), which at the moment are distracting. Second, some kind of statistical metric

could be used to indicate that there is indeed a relationship between AAI, ΔBT and SO2. If the relationship is not linear then maybe some kind of curve fit (exponential for lower right panel?) will help the reader interpret the relationships.

- *'HIMAWARI VADUGS ΔBTs' should read 'HIMAWARI ΔBTs'.*

- *The point of the comparison between ΔBT on the one hand and the AAI and SO2 on the other hand is to check if there was any relation between both. Based on passed literature showing that the magnitude of ΔBT is not related to ash optical depth (Figure 2 of [Prata & Prata, 2012] provides a nice illustration of this point), the "naive" assumption would be that ΔBT then should also not show a clear relation with the AAI and SO2. Which is doesn't, as the largest AAI and SO2 values occur for smaller ΔBTs. This is in a way a "negative" result, lack of a relationship between two parameters, but we felt that it was important to show this, as it highlights our point that there is added value in combining IR ΔBT with UV/VIS AAI and SO2.*

  *We added the following sentence to this section:*

  "Overall, we find little evidence of large AAI values and large $SO_2$values associated with large ΔBTs. Rather, their relation is complex."

- *In addition, we also added the following to the discussion & conclusion section (added section underlined here):*

  "… synergistically combining different satellite data products like the AAI and $SO_2$. Furthermore, ΔBT appears not to be a good indicator of either large AAI values or large $SO_2$ columns. This is not surprising as ΔBT is not a good indicator for ash optical depth [*e.g.* Prata and Prata, 2012; Pavolonis et al., 2016]. Our results therefore highlight that there is added value in combining IR ΔBT with UV/VIS AAI and $SO_2$. Satellite measurements … "

- *The solid and dotted contours denote outline of TROPOMI > 10 DU $SO_2$ columns and TROPOMI AAI > 0 value. This information is provided in the figure caption of figure 4, which the figure caption of figure 5 refers to. We added the same explanation to the caption of figure 5.*

- *Figure 5 is modified with now only showing three panels (3x1 rather than 2x2), with in the spatial regridded ΔBTs only the ΔBTs within the $SO_2$ /AAI contours (see here below for explanation of the contours). The lower two plots have remained. Color bars have been added. The figure caption now reads:*

  "**Figure 5. (A)** HIMAWARI-8 ΔBTs for 19 February 2018 06:30 UTC (see also Figure 4) regridded to the TROPOMI measurement grid of that day, and correlations between the HIMAWARI-8 ΔBTs and TROPOMI **(B)** AAI and **(C)** $SO_2$. The solid and dotted contours denote outline of TROPOMI > 10 DU $SO_2$ columns and TROPOMI AAI > 0 value, as also shown in figure 4 and derived from Figure 1. The color coding of the dots in the AAI scatterplots is indicative of the corresponding $SO_2$ value (> 10 DU) , and the color coding in the $SO_2$ scatterplot is indicative of the AAI value (AAI > 2), see also the lower color bar. These color codings were added for qualitatively identifying possible relationships between ΔBT and AAI or $SO_2$ within the volcanic ash plume."

P7L224: 'The larger ΔBT are also associated with optically more dense clouds (see VIIRS imagery in the SI and comparison of TROPOMI with CALIPSO).' - This statement needs to be further clarified. It's

not physically possible for an optically thick cloud to have a large ΔBT in the infrared. When clouds become optically thick they behave as grey bodies (little spectral variation across thermal infrared wavelengths) and so a difference in brightness temperature between 11 and 12 micron should be close to zero. However, I think what the authors are observing is a relationship between high reflectance at visible wavelengths (white clouds in VIIRS imagery) and large ΔBTs, but it's not clear in the way that it's stated.

- *We agree, the comparison with ΔBT and $SO_2$ /AAI requires careful wording. The main points we want to make is that the relationship between all three parameters is complex, and that we find indications of possible shielding effects (thick ash/ice shielding part of the underlying ash/$SO_2$ column). We modified this section as follows:*

"Figure 5 shows a comparison of TROPOMI AAI and $SO_2$ data with regridded HIMAWARI-8 ΔBTs (upper left plot). When focusing on AAI and $SO_2$ values, it appears that larger ΔBT values occur for smaller AAI values (< 2) and $SO_2$ columns (< 20 DU). The largest positive ΔBT are associated with optically thicker/less transparent water and ice clouds (see also VIIRS imagery in the SI and comparison of TROPOMI with CALIPSO). The lack of larger AAI and $SO_2$ values for larger positive ΔBT values therefore may reflect some kind of shielding of the volcanic ash and $SO_2$ by the iced upper levels of the volcanic ash cloud. $SO_2$ may have been converted into sulphate as the SO depletion rate (e-folding time), which, although uncertain, has been estimated to be as small as 5-30 minutes [Oppenheimer et al., 1998; McGonigle et al., 2004], scavenged by ice [Rose et al., 2000], or via ice nucleation of volcanic ash particles [Durant et al., 2008]. For negative ΔBTs – indicative of volcanic ash – we also find little evidence of a distinctive relation between either the AAI and $SO_2$ with ΔBTs. This may similarly reflect a shielding effect, as the largest ΔBTs do not occur for the largest aerosol concentrations [*e.g.* Prata and Prata, 2012; Pavolonis et al., 2016]."

P8L225-227: This could be due to scavenging of SO2 by ice (Rose et al., 2000). It could also be due to ice nucleation of volcanic ash particles (Durant et al., 2008). In terms of the conversion of SO2 to sulphate, is there a reference that could be added here? i.e. how long does it typically take for SO2 to convert to sulphate in the upper troposphere? And does this conversion rate make sense given the time of observation and time since eruption?

- *Mechanisms and references included in the text (see previous question for text modifications). The SO2-to-sulphate conversion time scale (e-folding time) is highly uncertain but can has been estimated as small as 5-30 minutes. Both remarks – including references – have also been added (again, see previous question for the new text).*

P8L245-248: 'Comparison with geostationary IR volcanic ash height' - Which retrieval is this statement referring to? Is this the VADUGS volcanic ash cloud height retrieval? It's the comparison with CALIOP that demonstrates TROPOMI height algorithms may underestimate heights for semi-transparent ash clouds. Please clarify this.

- *Should be CALIOP, not geostationary IR. This has been changed.*

P8L251-252: The 'shielding' effect - This is rather speculative and could be due to a number of different reasons (see previous comments on P8L225-227). Also, is this shielding of SO2 or ash or both? I think to substantiate this claim, evidence of SO2/ash existing underneath the cloud-top should be provided.

- *As explained, the lack of a distinctive relation between ΔBTs (both positive and negative) on the one hand and AAI and SO2 on the other hand we interpret as indications for shielding effects, see further the answer to P7L224. There does not exist other data to substantiate this claim. However, with the mentioning of other processes like SO2 conversion, depletion, scavenging, and nucleation, it should be clear to the reader that the analysis is by far definitive with regarding to the presence of a shielding effect.*

P9L257-258: 'the retrieval algorithm' - which retrieval algorithm is being referred to here? Please clarify.

- *Should read "IR volcanic ash retrieval algorithms", changed accordingly.*

P9L266-268: 'TROPOMI cloud heights can be used for determining aerosol heights for AAI values greater than 4' - How was this conclusion reached? What is the significance of AAI > 4. As stated in the previous sentence, the TROPOMI cloud heights do not perform well for semi-transparent clouds regardless of their AAI value. This statement requires further clarification. Also 'column values > 1 DU' is TROPOMI's signal-to-noise really this good? Please provide a reference.

- *preamble: although the AAI is not a quantitative parameter, in general there is – all else being equal - a clear correlation between the AAI and the AOD. The problem here is the "all else being equal": the AAI value depends on many parameters, like aerosol type, aerosol height, solar zenith angle, viewing angle, albedo below the aerosol layer (which includes clouds), and some more [de Graaf et al., 2005].*

  *A large AAI value will – generally speaking – indicate thick aerosol layers. Both user experience and theoretical calculations suggest – again, generally speaking – that AAI values larger than 2 (two) indicate significant amounts of aerosols. To be on the safe side and because the main interest is also to have non-transparent aerosol layers, we increased the AAI threshold to 4 (four). According to de Graaf et al. [2005], for aerosol layers with AAI values > 4, the AOD generally will be (much) larger than one (AOD of 1 = 1/e or 36.7% of the light is not scattered by the aerosol layer). At such scattering levels, aerosol layers become opaque.*

  *We added the following sentence:*

  "This AAI threshold value of 4 may be conservative but ensures that the aerosol layer very likely is opaque, as generally the associated aerosol optical depth will be (much) larger than on [de Graaf et al., 2005]."

- *TROPOMI SO2 accuracy is estimated to be at least 1 DU, but likely even better, see Theys et al. [2017]. Reference added.*

P11L323-325: Please fix reference formatting here. Also link provided to Stein Zweers (2016) results in a 'Page not found' error.

- *changed (document recently officially accepted by ESA, so the filename changed)*

P16L411: Check style for figure labels e.g. 'A+E' should be '(a) and (e)'.

- *checked and updated*

P19L415: VADUGS cloud heights are on the right column of Fig. 4 not left and the ΔBTs are on the left.

- *changed*

P19L427: Change 'derived' to 'shown'.

- *changed*

P19L427: What is the ΔBT bias correction? This needs to be explained and defined in the manuscript.

- *This refers to a typical value of the atmospheric correction of geostationary infrared brightness temperatures for this part of the world, but I do not know why this remark – stemming from an email exchange - has managed to seep into the paper. This can be removed.*

---

## Author Comment (AC2) · 6 Feb 2020

**Response to referee #2**

P3, L66: Missing word '(down to 2.5x7km2) of SO2'

- *changed*

P3, L70. The term "volcanic clouds" is mis-leading, since it can refer to clouds of ash, particles, trace-gases. Here you are clearly referring to volcanic ash clouds. You use this term very often throughout the text. Please describe at each occasion what you mean.

- *systematically changed "volcanic cloud" to "volcanic ash", "volcanic ash plumes" or "volcanic ash and $SO_2$ plumes", depending on the context.*

P3, L84: Calipso -> CALIPSO P3, L85. Maybe add a short explanation about what the 'A-train constellation' is.

- *changed to:*

  "CALIPSO is part of the A-train constellation, which consists of several Earth-observing satellites that closely follow one another, crossing the equator in an ascending (northbound) direction at about 1:30 PM local solar time, within seconds to minutes of each other along the same or a very similar orbital "track"."

P3, L85. It is "13:30h local time"

- *changed*

P4,L118: Add here that the SO2 product provides four different SO2 VCDs for different SO2 vertical profile shapes, since they are not known at the time of the measurement. For the rest of the paper it would be also good to know, which SO2 VCD you have chosen. Here you might also refer to the paper of Hedelt et al. 2019, who has also studied the Sinabung eruption and retrieved SO2 plume heights for this.

- *we added the following:*

  "The TROPOMI $SO_2$ data product provides four different $SO_2$ VCDs for different $SO_2$ vertical profile shapes, since they are not known at the time of the measurement. For this paper, we use the standard $SO_2$ VCD product."

- *We also added to section 3.1:*

  "These heights are consistent with results the recently introduced new TROPOMI $SO_2$ height data product [Hedelt et al., 2019]."

- *we also added to the summary and discussion in relation to the use of the standard SO2 data product rather than for example the 15 km SO2 data product the following:*

  "Also note that it could be argued that it would be better to use the TROPOMI SO2 15 km data product, as 15 km is more consistent with the volcanic plume height. However, this 15 km data product assumes a "nice and tidy" $SO_2$ plume without any contamination, let alone the complexity of a fresh, optically very thick eruption plume and the presence of condensed

water, in combination with indications of a shielding effect. Furthermore, the main focus of this paper is ash heights rather than $SO_2$, which is mostly used as a proxy for a volcanic plume, although investigating the accuracy and precision of satellite $SO_2$ VCD observations in fresh volcanic plumes would be valuable, in particular with soon to be launched geostationary hyperspectral satellites."

P4, L123: Add a reference for the O22CLD algorithm (either paper or ATBD)

- *reference to Veefkind et al. [2016] added*

P4, L129. Consider also adding information about the cloud fraction from OCRA

- *added the following:*

  "Note that TROPOMI operational cloud fractions are derived from the OCRA algorithm [Loyola et al., 2018]."

P5, Sect. 2.5: I suggest to add more information on CALIOP, references and a description of what the 'attenuated backscatter imagery' displays, i.e. what it is sensitive to, etc. I also propose to also add the VFM, which shows the type of absorption feature as well as the BTD which gives information about the type of absorption.

- *we added the following to section 1.5*

  "The TAB signal strength is color coded such that blues correspond to molecular scattering and weak aerosol scattering, aerosols generally show up as yellow/red/orange. Stronger cloud signals are plotted in gray scales, while weaker cloud returns are similar in strength to strong aerosol returns and coded in yellows and reds. The TAB in sensitive to atmospheric particles: both water and ice droplets as well as various types of aerosols."

- *with regard to the VFM, we added halfway section 3.3 a reference to Hedelt et al. [2019] who present and analyze the VFM for the same CALIPSO orbit, and conclude that the volcanic cloud "contains high concentrations of water droplets".*

- *we further investigated the corresponding CALIPSO DBT (10-12 micron; see below), but found the small swath of the CALIPSO DBT difficult to interpret without the context provided by for example HIMAWARI-8 time series in conjunction with the various TROPOMI data products. Since we already extensively show and discuss HIMAWARI-8 DBT, there does not appear to be much added value in the small-swath CALIPSO DBT plot.*

[Figure]

P6, L163 You write the 'extend of the volcanic plume', but by means of what? SO2 VCD or AAI or? Please specify.

- *changed to:*

  "The AAI and $SO_2$ contours agree well with the cloud structure associated with the volcanic plume , …"

P6, L166: Here it would be interesting to see what is the TROPOMI OCRA cloud fraction.

- *the FRESCO cloud fraction (figure 1, panel [D]) shows that there is little cloud fraction structure resembling the volcanic plume, which is also the reason we only continue to use cloud heights/pressures.*

P6, L170 Please describe the 'clear differences' between FRESCO and ROCINN

- *added the following:*

  "Differences between FRESCO and ROCINN for the volcanic plume are small, most notably the lack of saturated pixels in ROCINN (greys in FRESCO), possible due to the neural network filling in the gaps with nearby cloud information or interpolating between cloud pixels."

P6, L183 I suggest to rephrase the sentence, since the CALIOP data only shows an attenuation by clouds. As you write later on, there is no *CLEAR* detection of an ash layer

- *currently these layers are characterized in the text as "cloud/ash" layers to reflect the ambiguity about CALIPSO not always being able to discriminate between clouds particles and aerosols. We honestly think this should be sufficient. Hedelt et al. [2019] does not even make this reservation. Instead, they directly conclude that this must be and ash/aerosol layer – only supported by the VFM identification of a few aerosol pixels, where most pixels suggest it is a cloud (which they then attribute to water/ice within the volcanic plume).*

P6, L187: Add the CALIPSO overpass time here, such that the reader gets an idea about the overpass time difference btw TROPOMI & CALIPSO

- *added the following to the first paragraph of the section:*

  "The CALIOP overpass time of this area is between 07:09:56 and 07:11:26 UTC, the TROPOMI overpass time is between 06:24:23 and 06:26:00 UTC, a time difference of approximately 45 minutes."

P7, L193. The VFM classifies the volcanic cloud as 'cloud' and sometimes 'ash'. This is because fresh volcanic plumes are typically rich in water vapor (especially for tropical eruptions). The volcanic clouds also contain high concentrations of water droplets. Therefore, the classification in the CALIPSO VFM sometimes fails to pick up the volcanic ash or sulfate aerosol because of competing clouds. Another interesting feature which could be analyzed in this paper is the brightness temperature difference from CALIPSO which clearly shows the ash in the data

- *as shown above, although there are clear BTD signatures in the CALIPSO imager data CALIPSO, there appears little added value to what HIMAWARI shows.*

- *We added the following sentence:*

  "The lack of aerosol masking in the feature mask most likely is related to liquid water or ice contaminating the volcanic ash [Hedelt et al., 2019]."

P7 L214-216. The description of the BTD should appear in Sect. 2.4

- *moved to section 2.4*

P8 L237: TROPOMI was launched in 2017. Given that we now have 2019, I wouldn't call it 'recently launched'.

- *removed*

P8 L225-226 Since the ash and SO2 cloud are co-located there is certainly also an effect of the ash on the SO2 VCD retrieval (and not only shielding). Might be interesting to study the effect of ash on the SO2 VCD

- *we added a recommendation of investigating SO2 retrievals in fresh volcanic plumes to the summary and discussion section*

Figures

Figure 1 is clearly overloaded. Although it is interesting to see all results in one single figure, it is really hard to understand all plots. I would suggest to break down the figure into several figures (i.e. show FRESCO/OCRA/O22CLD cloud heights separately in a figure, as well as FRESCO/OCRA CF and SO2/ AAI) before showing combined plots.

- *we have split figure 1 in two separate ones (1A and 1B, both with our panels). 1A shows the ROCINN CP and the FRESCO CTH, with and without the SO2 and AAI overlaid. This highlights the correspondence between the AAI, SO2, and the cloud height/pressure. 1B shows the FRESCO cloud fraction and apparent scene pressure, and the O22CLD cloud height and apparent scene pressure. This illustrates that the FRESCO cloud fraction does not show spatial structures consistent with the volcanic plume, but that the FRESCO scene pressure and the O22CLD cloud height and scene pressure also show similar structures. In all plots the same SO2 and AAI contours are added for visual guidance. Combined, 1A and 1B provide a consistent view of the various cloud altitude products.*

Figure 2: Clearly a colorbar for the attenuation backscatter is missing. Also please add the AAI colorbar to this figure and not only refer to it. Please consider to show also the VFM, showing some ash classification in the cloud

- *see also our response to Referee #1, P5, Sect. 2.5:*

  *with regard to the VFM, we added halfway section 3.3 a reference to Hedelt et al. [2019] who present and analyze the VFM for the same CALIPSO orbit, and conclude that the volcanic cloud "contains high concentrations of water droplets".*

  *This whole section now reads:*

  *"Note that CALIOP's own feature mask does not identify hardly any of these backscatter signals as aerosol (for CALIOP v4.10 an occasional cloud pixel is flagged as aerosol, see Hedelt et al., [2019]): the high-altitude structures are flagged as regular clouds, and the below-cloud structure as "totally attenuated", even though clearly the attenuation is not complete. The lack of aerosol masking in the feature mask most likely is related to liquid water or ice contaminating the volcanic ash [Hedelt et al., 2019]."*

- *attenuation backscatter color bar has been added to figure 2*

Figure5: I would rearrange the plots and show the HIMAWARI vs AAI plots next to each other and clearly indicate the SO2 threshold in the plot title. A color bar for each sub plot would also be very helpful. Have you tried to display these results in a 3D scatter plot, with x=SO2, y=AAI and z=BTD? Furthermore, why did you show the scatterplots for AAI < -0.25 – they are not part of the volcanic ash cloud, as your AAI contours also indicate. For the SO2 plot I suggest using a logarithmic scale. I also suggest showing horizontal/vertical lines at x=0 and y=0

  *in conjunction with comments by Referee #1, we modified Figure 5 which now only shows three panels (3x1 rather than 2x2), with in the spatial regridded ΔBTs only the ΔBTs within the SO$_2$ /AAI contours (see here below for explanation of the contours). The lower two plots have remained, but with only data from within the SO2+AAI contours.*

*There remain some AAI values < 0.0 because we combine the SO2 and AAI contours to define the eruption plume outline, the AAI and SO2 contours do not exactly overlap, and the contours - by construction – represent a smooth outline which not exactly follows the SO2 and AAI threshold values. Nevertheless, by only using data within the contours the number of pixels with AAI < 0.0 is limited.*

*The negative AAI values are found at the eruption plume edge and in the region where the DBT values are positive, indicative of snow/ice rather than volcanic ash. Negative AAI values can be related to water/ice clouds, which can produce zero or negative AAI values [de Graaf et al., 2005; 10.1029/2004JD005178], which is consistent with the presence of negative AAI values despite this still clearly being part of the volcanic plume as derived from following the volcanic plume development and dispersion in the HIMAWARI-8 data.*

*A color bar for plot (B) and (C) has been added below the plot, which, combined with the rearrangement from 2x2 to 3x1 should help make the plot easier to view/read/interpret.*

*The figure caption now reads:*

*"**Figure 5. (A)** HIMAWARI-8 ΔBTs for 19 February 2018 06:30 UTC (see also Figure 4) regridded to the TROPOMI measurement grid of that day, and correlations between the HIMAWARI-8 ΔBTs and TROPOMI **(B)** AAI and **(C)** SO$_2$. The solid and dotted contours denote outline of TROPOMI > 10 DU SO$_2$ columns and TROPOMI AAI > 0 value, as also shown in figure 4 and derived from Figure 1. The color coding of the dots in the AAI scatterplots is indicative of the corresponding SO$_2$ value (> 10 DU) , and the color coding in the SO$_2$ scatterplot is indicative of the AAI value (AAI > 2), see also the lower color bar. These color codings were added for qualitatively identifying possible relationships between ΔBT and AAI or SO$_2$ within the volcanic ash plume."*

Figures S1, S2a, S2b: Would it be possible to indicate the location of the volcano on the maps?

- *volcano location indicated with red triangle*

Figures S2: Would it be possible to add the AAI/SO2 contours to the maps?

- *contours of SO2 and AAI as in figure 1 added to figures S2A/S2B*

---

## Referee Report (RR1)

**Review round 2 of 'Analysis of properties of the 19 February 2018 volcanic eruption of Mount Sinabung in S5P/TROPOMI and Himawari satellite data' by de Laat et al.**

Thank you for addressing my initial review comments. The majority of my comments have been adequately addressed and I believe the paper should be published after addressing some technical corrections attached below.

**Technical corrections**

Why is 'HIMAWARI-8' capitalised? Why not just 'Himawari-8'? The JMA themselves do not capitalise it (see Bessho et al., 2016). This is a minor grammatical issue, but easy to fix.

P3L87: 'CALIPSO is part of the A-train constellation...' - At the time of the Sinabung eruption this was true. But CALIPSO is now part of the 'C-train' - might be worth mentioning here. See here for more info: https://atrain.nasa.gov/.

P5L149: What does 'nontrivial' mean exactly? Theory predicts positive BTD values for liquid water droplets and ice particles. I suggest deleting 'nontrivial'.

P6L163-164: Suggest replacing 'ash plumes' with 'components'.

P9L261: 'as the largest $\Delta$BTs do not occur for the largest aerosol concentrations'. I would re-word this slightly to 'as large negative $\Delta$BTs do not necessarily occur for large aerosol concentrations'.

P10L302: 'Furthermore, $\Delta$BT appears not to be a good indicator of either large AAI values or large $SO_2$ columns.' I would word this more carefully. I suggest changing to 'Furthermore, for the present case study, large negative $\Delta$BT values appear not to be a good indicator of large AAI values (or large $SO_2$ columns)'.

P10L302-303: 'This is not surprising as $\Delta$BT is not a good indicator for ash optical depth [e.g. Prata and Prata, 2012; Pavolonis et al., 2016].'. This sentence is not correct because it implies that $\Delta$BT is not related to ash optical depth. Figure 2 of Prata and Prata (2012) shows the relationship between $\Delta$BT, 11 $\mu$m optical depth and effective radius. The three variables depend on each other. I suggest revising the sentence to 'This is not surprising as highly negative $\Delta$BT values do not necessarily indicate high ash optical depths [e.g. Prata and Prata, 2012; Pavolonis et al., 2016].'.

P10L313-314: 'as generally the associated aerosol optical depth will be (much) larger than on [de Graaf et al., 2005]..'. This sentence doesn't make sense. 'than on' what?

P10L314: Replace 'The combination of UV/VIS cloud heights' with 'For the combination of UV/VIS cloud heights'.

**References**

Bessho, K., Date, K., Hayashi, M., Ikeda, A., Imai, T., Inoue, H., Kumagai, Y., Miyakawa, T., Murata, H., Ohno, T., Okuyama, A., Oyama, R., Sasaki, Y., Shimazu, Y., Shimoji, K., Sumida, Y., Suzuki, M., Taniguchi, H., Tsuchiyama, H., Uesawa, D., Yokota, H. and Yoshida, R.: An Introduction to Himawari-8/9Japans New-Generation Geostationary Meteorological Satellites, Journal of the Meteorological Society of Japan. Ser. II, 94(2), 151183, doi:10.2151/jmsj.2016-009, 2016.

---

## Author Response (AR2)

P4, L102 UV abbreviation first appears here, but could already appear P2 L65

changed

P5 L139 UV abbreviation never defined before, could be done on P2 L65

See previous, changed. We also double checked all abbreviations to as much as possible consistently define abbreviations at the first instance. Note that generally earth observation is an abbreviation-rich part of science, hence the lengthy glossary

P5 L134 Missing word: It is expected that *IN* the coming years a surface albedo...

changed

P5 L165 Missing word: These heights are consistent with results *OF* the recently...

changed

P6 L182-184 There is some confusion and a missing word in this sentence:
...in the FRESCO cloud height and ROCINN cloud *PRESSURE* ...
...as well as in the ROCINN and O22CLD scene pressures... ROCINN is not shown in Fig1b, but I think here you mean FRESCO

Correct, changed, and apologies, the modifications of figure 1 caused some textual inconsistencies that escaped our attention.

P6, L187: According to Fig3, FRESCO and ROCINN cloud heights differ by up to 5km in the center of the plume. So the sentence 'cloud heights are rather similar' is not really correct... Please rephrase

P6 L187: The sentence 'However, there are also clear differences.' does not make sense here, because it refers to the sentence before about FRESCO and ROCINN. However, the next sentence is related to differences between O22CLD and FRESCO. If you want to keep this statement here, you could move the last sentence of the paragraph (P7 L194-197) here, which refers to FRESCO/ROCINN differences... Btw - this last sentence is however (strictly speaking) not correct, when you look at Fig 3 (see my above comment)

Changed to "The FRESCO and ROCINN cloud heights both consistently indicate cloud heights of 10 km or higher, the O22CLD cloud heights also reach 10 km but for fewer pixels and in general FRESCO and ROCINN cloud heights are higher than the O22CLD cloud heights (figure 1B)."

We also changed at the end of the section "Differences between FRESCO and ROCINN for the volcanic plume are smaller" with "… appear less striking". We added to the end of the section the following sentence: "However, it appears that FRESCO cloud heights are higher for the northern half of the ash plume. FRESCO cloud heights exceed 12.5 km, which is approximately 200 hPa, ROCINN cloud pressure does not appear to exceed 200 hPa."
This then is a natural coupling with the comparison with CALIPSO later on, which shows no ROCINN heights beyond 200 hPa.

We added then the following to the discussion (section 4).

"However, there is a difference between FRESCO and ROCINN for very high FRESCO heights (> 12.5 km or approximately 200 hPa). This might indicate that the ROCINN neural network may not be that sufficiently trained on clouds beyond 12 km or 200 hPa. "

P10 L314: Incomplete sentence: ...as generally the associated aerosol optical depth will be (much) larger than on [de Graaf et al., 2005].

Changed to "as generally the associated aerosol optical depth will be (very) large"

P11 No reference to Hedelt et al. 2019 in the References
added

Figure 5b: Figure title and caption are not correct. In the title and text it is written that the data is color-coded for SO2>20DU, which is a light-red color according to the SO2 color bar. In the plot however several dark red points appear, indicating much lower SO2 values around 0-10DU. Please correct.

Correct, the main body text should read ">10 DU". Also note that the main body text still referred to the upper left plot of figure 5, which now is panel A of figure 5. This has also been corrected.

Review round 2 of `Analysis of properties of the 19 February 2018 volcanic eruption of Mount Sinabung in S5P/TROPOMI and Himawari satellite data' by de Laat et al.

Thank you for addressing my initial review comments. The majority of my comments have been adequately addressed and I believe the paper should be published after addressing some technical corrections attached below.

Technical corrections

Why is `HIMAWARI-8' capitalised? Why not just `Himawari-8'? The JMA them- selves do not capitalise it (see Bessho et al., 2016). This is a minor grammatical issue, but easy to _x.

Apologies, occupational hazard: generally names of satellites are abbreviations and natural tendency is thus to write all of them in capitals without thinking. But of course the referee is correct, Himawari is not an abbreviation. → changed

P3L87: `CALIPSO is part of the A-train constellation...' - At the time of the Sinabung eruption this was true. But CALIPSO is now part of the `C-train' – might be worth mentioning here. See here for more info: https://atrain.nasa.gov/.

Added to the introduction: "Note that after an orbital maneuver in September 2918, CALIPSO is not part of the A-train constellation anymore."

P5L149: What does `nontrivial' mean exactly? Theory predicts positive BTD values for liquid water droplets and ice particles. I suggest deleting `nontrivial'.

The use of "nontrivial" directly refers to Pavolonis [2006], who state that:

"Very thick ash clouds or ash plumes with nontrivial liquid water or ice contents will generally have a positive BTD[11, 12]."

… although Pavolonis [2006] does not explain what is meant by "nontrivial", rather referring to Prata et al. [2001 ; Remote Sensing of Environment 78 (2001) 341–346], who discuss examples of failed split-channel-technique detection of volcanic ash in tropical regions due to the presence of liquid water vapor or ice but without use of the expression "nontrivial".

It is well established that the presence of condensed water/ice can compromise the IR split-channel detection of volcanic ash, especially in the moist tropics. Different mechanisms and processes have been identified or proposed that can result in difficulties with IR split-channel volcanic ash detection (see Prata et al. [2001]).

But agreed, we deleted "nontrivial"

P6L163-164: Suggest replacing `ash plumes' with `components'.

changed

P9L261: `as the largest _BTs do not occur for the largest aerosol concentrations'. I would re-word this slightly to `as large negative _BTs do not necessarily occur for large aerosol concentrations'.

Agreed, but we suggest to modify it as follows, because looking at the theoretical calculation the largest ΔBTs as can be found in the referenced papers – and thus the split-channel technique for detecting volcanic ash and aerosols – works best for optically somewhat thinner clouds.

"as the largest aerosol concentrations are not associated with the largest possible ΔBTs"

P10L302: `Furthermore, _BT appears not to be a good indicator of either large AAI values or large SO2 columns.' I would word this more carefully. I suggest changing to `Furthermore, for the present case study, large negative _BT values appear not to be a good indicator of large AAI values (or large SO2 columns)'.

changed

P10L302-303: `This is not surprising as _BT is not a good indicator for ash optical depth [e.g. Prata and Prata, 2012; Pavolonis et al., 2016].'. This sentence is not correct because it implies that _BT is not related to ash optical depth. Figure 2 of Prata and Prata (2012) shows the relationship between _BT, 11 _m optical depth and e_ective radius. The three variables depend on each other. I suggest revising the sentence to `This is not surprising as highly negative _BT values do not necessarily indicate high ash optical depths [e.g. Prata and Prata, 2012; Pavolonis et al., 2016].'.

changed to "This is not surprising as highly negative ΔBTs do not necessarily indicate large ash optical depth values"

P10L313-314: `as generally the associated aerosol optical depth will be (much) larger than on [de Graaf et al., 2005]..'. This sentence doesn't make sense. `than on' what?

Typo, changed to "as generally the associated aerosol optical depth will be (very) large [de Graaf et al., 2005]".

Note that de Graaf et al. [2005] show a plot of how AAI values increase with increasing aerosol optical depth. However, as the AAI value also depends on other parameters like aerosol height and the Angstrom coefficient, the relation between AAI and AOD cannot be uniquely defined. Hence the use of "associated", and much like ΔBTs do not uniquely depend on AOD as well.

P10L314: Replace `The combination of UV/VIS cloud heights' with `For the combination of UV/VIS cloud heights'.

changed

References
Bessho, K., Date, K., Hayashi, M., Ikeda, A., Imai, T., Inoue, H., Kumagai, Y., Miyakawa, T., Murata, H., Ohno, T., Okuyama, A., Oyama, R., Sasaki, Y., Shimazu, Y., Shimoji, K., Sumida, Y., Suzuki, M., Taniguchi, H., Tsuchiyama, H., Uesawa, D., Yokota, H. and Yoshida, R.: An Introduction to Himawari-8/9 Japans New-Generation Geostationary Meteorological Satellites, Journal of the Meteorological Society of Japan. Ser. II, 94(2), 151183, doi:10.2151/jmsj.2016-009, 2016.

[revised manuscript text omitted]